# Optimality of Message-Passing Architectures for Sparse Graphs

**Aseem Baranwal**
David R. Cheriton School of Computer Science
University of Waterloo, Waterloo, Canada
`aseem.baranwal@uwaterloo.ca`

**Kimon Fountoulakis**
David R. Cheriton School of Computer Science
University of Waterloo, Waterloo, Canada
`kimon.fountoulakis@uwaterloo.ca`

**Aukosh Jagannath**
Department of Statistics and Actuarial Science,
Department of Applied Mathematics,
David R. Cheriton School of Computer Science
University of Waterloo, Waterloo, Canada
`a.jagannath@uwaterloo.ca`

## Abstract

We study the node classification problem on feature-decorated graphs in the sparse setting, i.e., when the expected degree of a node is $O(1)$ in the number of nodes, in the fixed-dimensional asymptotic regime, i.e., the dimension of the feature data is fixed while the number of nodes is large. Such graphs are typically known to be locally tree-like. We introduce a notion of Bayes optimality for node classification tasks, called asymptotic local Bayes optimality, and compute the optimal classifier according to this criterion for a fairly general statistical data model with arbitrary distributions of the node features and edge connectivity. The optimal classifier is implementable using a message-passing graph neural network architecture. We then compute the generalization error of this classifier and compare its performance against existing learning methods theoretically on a well-studied statistical model with naturally identifiable signal-to-noise ratios (SNRs) in the data. We find that the optimal message-passing architecture interpolates between a standard MLP in the regime of low graph signal and a typical convolution in the regime of high graph signal. Furthermore, we prove a corresponding non-asymptotic result.

## 1 Introduction

Graph Neural Networks (GNNs) have rapidly emerged as a powerful tool for learning on graph-structured data, where along with features of the entities, there also exists a relational structure among them. They have found numerous applications to a wide range of domains such as social networks [Backstrom and Leskovec, 2011], recommendation systems [Ying et al., 2018, Hao et al., 2020], chip design [Mirhoseini et al., 2021], bioinformatics [Scarselli et al., 2009, Zhang et al., 2021], computer vision [Monti et al., 2017], quantum chemistry [Gilmer et al., 2017], statistical physics [Battaglia et al., 2016, Bapst et al., 2020], and financial forensics [Zhang et al., 2017, Weber et al., 2019]. Most of the success with these applications has been possible due to the advent of the

37th Conference on Neural Information Processing Systems (NeurIPS 2023).

message-passing paradigm in GNNs, however, designing optimal GNN architectures for such a wide variety of applications still remains a challenging task.

In this work, we are interested in the node classification problem on very sparse feature-decorated graphs that are locally tree-like. We focus on the regime where the dimension of the node features is fixed and the number of nodes is large. Our motivation for considering this regime is that many major benchmark datasets for node classification appear to scale in this fashion. For example, in the popular Open Graph Benchmark collection [Hu et al., 2020], the medium and large-scale node-property prediction datasets have roughly $10^6$ nodes (`ogbn-products`, `ogbn-mag`) to about $10^8$ nodes (`ogbn-papers100M`), each with roughly $10^2$ features. Graphs with such properties exist naturally in social, informational and biological networks; for motivational examples, see Stelzl et al. [2005], Adcock et al. [2013]. We present a precise definition of optimality for node classification tasks on locally tree-like graphs in this scaling regime and compute the optimal classifier according to this definition, for a multi-class statistical data model where node features can have arbitrary continuous or discrete distributions. Subsequently, we show that a message-passing GNN architecture is able to realize the optimal classifier. Furthermore, we provide a theoretical analysis, comparing the generalization error of the optimal classifier with other architectures like GCN and simple MLPs. Our results support a recent work [Veličković, 2022] in the context of classification on sparse graphs. In particular, we show that when node features are accompanied by sparse graphical side information, message-passing graph neural networks are able to realize the optimal classification scheme, and as such, there does not exist a better architecture beyond the message-passing paradigm.

**Related Work.** There has been a tremendous amount of work on GNN architecture design, where the most popular designs are based on a convolutional architecture, with each layer of the neural network performing a weighted convolution (averaging) operation with immediate neighbours, e.g., graph convolutional networks (GCN) [Kipf and Welling, 2017, Chen et al., 2020] or graph attention networks (GAT) [Veličković et al., 2018]. These architectures are known to have several limitations regarding their expressive power (see, for e.g., Li et al. [2018], Oono and Suzuki [2020], Balcilar et al. [2021], Xu et al. [2021], Keriven [2022]).

An interesting line of research consists of both theoretical and empirical works that attempt to address these limitations by developing an understanding of GNN architectures within the scope of message-passing [Rong et al., 2020, Liu et al., 2022, Maskey et al., 2022], as well as beyond it [Maron et al., 2019, Murphy et al., 2019, Chen et al., 2019]. For example, Xu et al. [2018] propose an architecture with a technique called skip-connections, that flexibly leverages different ranges of neighbourhoods for each node to enable structure-awareness in node representations, Chen et al. [2020] propose a modification of the vanilla GCN with an initial residual that effectively relieves the problem of oversmoothing [Oono and Suzuki, 2020], and Keriven et al. [2021] study the universality of structural GNNs in the large random graph limit. However, this area of research still lacks a clear understanding of optimality in the context of graph learning problems, making it hard to design architectures for which a well-defined notion of optimality can be theoretically justified.

Several works have studied traditional message-passing GNN architectures like GCN and GAT using the binary contextual stochastic block model, see for example, Baranwal et al. [2021], Chien et al. [2022], Fountoulakis et al. [2022a,b], Javaloy et al. [2022], Baranwal et al. [2023]. These analyses rely heavily on two assumptions: first, the graph is not too sparse, i.e., for a graph with $n$ nodes, the expected degree of a node is $\Omega_n(\log^2 n/n)$, and second, the node features are modelled as a Gaussian mixture. The work by Wei et al. [2022] is of particular interest to us, where the authors take a Bayesian inference perspective to investigate the functions of non-linearity in GNNs for binary node classification. They characterize the max-a-posterior estimation of a node label given the features of itself and its immediate neighbours. A similar perspective to that of Wei et al. [2022] is discussed in Gosch et al. [2023], where the latter authors derive insights into the robustness-accuracy trade-off in GNNs for node classification. In contrast to these inspiring works, we study the highly sparse regime where the expected degree of a node is $O_n(1)$ and consider nodes beyond the immediate neighbours, at any fixed distance. (In fact, our non-asymptotic results allow distances of order $c \log n$ for small enough $c > 0$, see Section 3.5 below.) Furthermore, our main result holds for a general multi-class statistical model with arbitrary continuous or discrete feature distributions and arbitrary edge-connectivity probabilities between all pairs of classes.

**Our Contributions.** In this paper, we use a multi-class statistical model with arbitrary node features and edge-connectivity profiles among all pairs of classes to study the node classification problem in the regime where the graph component of the data is very sparse, i.e., the expected degree is $O(1)$. The data model is described in Section 3.2. We state the following main results and findings:

1. We introduce a family of graph neural network architectures that are asymptotically (in the number of nodes $n \to \infty$) Bayes optimal in a local sense for a general multi-class data model with arbitrary feature distributions. The optimality is stated precisely in Theorem 1.

2. We analyze the architecture in the simpler two-class setting with Gaussian features, explicitly characterizing the generalization error in terms of the natural signal-to-noise ratio (SNR) in the data, and perform a comparative study against other learning methods analyzed using the same statistical model (Theorems 2 and 3). We find two key insights:

    - When the graph SNR is very low, the architecture reduces to a simple MLP that does not consider the graph, while if it is very high, our architecture reduces to a typical convolutional network that averages information from all nodes in the local neighbourhood. In the regime between the low and high SNRs, the architecture interpolates and performs better than both a simple MLP and a typical GCN.
    - If the information in the graph is larger than a threshold, then a simple convolution is able to perform better than all methods that do not utilize the graph. Not surprisingly, this threshold aligns with the Kesten-Stigum weak-recovery threshold for community detection in sparse networks [Massoulié, 2014, Mossel et al., 2018].

3. In the non-asymptotic setting with a fixed number of nodes, we show that even for a logarithmic depth, the neighbourhoods of an overwhelming fraction of nodes are tree-like with high probability. Subsequently, we show that the optimal classifier in the non-asymptotic setting obtains an error that is close to that incurred by the optimal classifier in the asymptotic setting. This is formalized in Theorem 4.

Let us end this section by reiterating that we work in the fixed-dimensional regime. It is natural to wonder as to the performance of this architecture as compared to optimal algorithms in the high-dimensional setting. We present a numerical comparison of our work to the AMP-BP algorithm from Deshpande et al. [2018] in low and high-dimensional settings in Appendix B.

## 2 Architecture

This section explains the design of our GNN architecture for node classification. We perform two modifications to existing message-passing architectures. First, we decouple the layers in the neural network from the neighbourhood radius in the message-passing framework. This style of decoupled architecture has previously been studied, see for example, Nikolentzos et al. [2020], Feng et al. [2022], Baranwal et al. [2023]. Second, we introduce a learnable parameter that models edge connectivity between each pair of classes and helps construct the messages to propagate. In the following, for any matrix $M$, we denote row $i$ of $M$ by $M_{i,:}$ and column $i$ of $M$ by $M_{:,i}$.

Before stating our architecture, we need the following additional notation and pre-processing. Let $\ell \geq 0$ and $L > 0$ be fixed integers. Let $C \geq 2$ be the number of classes. For given data $(\mathbf{A}, \mathbf{X})$ where $\mathbf{A} \in \mathbb{R}^{n \times n}$ is the adjacency matrix of an unweighted undirected graph, and $\mathbf{X} \in \mathbb{R}^{n \times d}$ is the node feature matrix, we perform a pre-computation on the graph to construct a tensor $\tilde{\mathbf{A}}$ as follows:

$$\tilde{\mathbf{A}}^{(k)} = f(\mathbf{A}^k) \wedge \left( \neg f \left( \sum_{m=0}^{k-1} \mathbf{A}^m \right) \right) \text{ for } k \in \{1, \ldots, \ell\},$$

where $f(M)$ for a matrix $M$ returns the entry-wise flattened matrix with $f(M)_{ij} = \mathbb{1}(M_{ij} > 0)$, and $(\wedge, \neg)$ denote the entry-wise bit-wise operators ('and', 'negation') respectively. Note here that $\tilde{\mathbf{A}}^{(k)}$ is an $n \times n$ binary matrix with $\tilde{\mathbf{A}}_{uv}^{(k)} = 1$ if and only if $v$ is present in the distance $k$ neighbourhood of $u$ but not within the distance $(k-1)$ neighbourhood. The idea behind this pre-processing step is the following: for each node $u$ and each $k \in [\ell]$, we want to divide the radius $\ell$ neighbourhood of $u$ into $\ell$ groups of nodes, where each group $k \in [\ell]$ consists of nodes that are within discovered the neighbourhood at each distance from a given node. $\tilde{\mathbf{A}}_{u,:}^{(k)}$ models a non-backtracking walk of length $k$ that considers new nodes in the distance-$k$ neighbourhood that were not discovered.

We can now define the graph neural network architecture as follows.

**Architecture 1.** Given input data $(\mathbf{A}, \mathbf{X})$ where $\mathbf{A} \in \{0,1\}^{n \times n}$ is the adjacency matrix and $\mathbf{X} \in \mathbb{R}^{n \times d}$ is the node feature matrix, define:

$$\mathbf{H}^{(0)} = \mathbf{X}, \qquad \mathbf{H}^{(l)} = \sigma_l(\mathbf{H}^{(l-1)}\mathbf{W}^{(l)} + \mathbf{1}_n \mathbf{b}^{(l)}) \text{ for } l \in [L],$$

$$\mathbf{Q} = \text{sigmoid}(\mathbf{Z}), \qquad \mathbf{M}_{u,i}^{(k)} = \max_{j \in [C]} \left\{ \mathbf{H}_{u,j}^{(L)} + \log(\mathbf{Q}_{i,j}^k) \right\} \text{ for } k \in [\ell], u \in [n], i \in [C].$$

Then the predicted label is given by $\hat{\mathbf{y}} = \{\hat{y}_u\}_{u \in [n]}$, where

$$\hat{y}_u = \underset{i \in [C]}{\text{argmax}} \left( \mathbf{H}_{u,c}^{(L)} + \sum_{k=1}^{\ell} \tilde{\mathbf{A}}_{u,:}^{(k)} \mathbf{M}_{:,i}^{(k)} \right).$$

Let us pause here to comment on the interpretation of the terms arising in this architecture. Here, $\mathbf{H}^{(L)}$ is viewed as the output of a simple $L$-layer MLP with $\{\sigma_l\}_{l \in [L]}$ being a set of non-linear functions. We have $(\mathbf{W}^{(l)}, \mathbf{b}^{(l)})_{l \in [L]}$ as the learnable parameters of this MLP, with suitable dimensions so that $\mathbf{H}^{(L)} \in \mathbb{R}^{n \times C}$. In addition, we introduce the learnable parameter $\mathbf{Z} \in \mathbb{R}^{C \times C}$ which is used to model edge connectivity among all pairs of classes. The quantity $\tilde{\mathbf{A}}_{u,:}^{(k)} \mathbf{M}_{:,i}^{(k)} = \sum_{v \in [n]} \tilde{\mathbf{A}}_{u,v}^{(k)} \mathbf{M}_{v,i}^{(k)}$ is viewed as the sum of messages $\mathbf{M}_{v,i}^{(k)}$ passed by all distance $k$ neighbours of node $u$.

Although Architecture 1 follows the style of convolutional architectures like GCN and GAT for collecting messages within a local neighbourhood, the novelty lies in the construction of the messages $\mathbf{M}$. Intuitively, $\mathbf{Q} = \text{sigmoid}(\mathbf{Z})$ learns the probabilities of edge connectivity between all pairs of classes so that $\mathbf{Q}^k$ models the probability of observing a distance $k$ path between a pair of nodes in two classes. To predict the label of node $u$, the messages from other nodes $v$ are constructed based on their features $\mathbf{X}_v$, their distance $k$ from node $u$, and the path probabilities $\mathbf{Q}^k$. We show in Theorem 1 that this architecture is in a sense (made precise in Definition 3.2) universally optimal among all node-classification schemes for sparse graphs. Our result thus aligns with the observations in Veličković [2022], showing that optimal neural network architectures for node classification on sparse graphs are implementable using the message-passing paradigm.

## 3 Theoretical Analysis and Discussion

In this section, we present a theoretical analysis of the message-passing GNN given in Architecture 1. We begin by defining a natural notion of optimality in our setting and show that among local learning methods on graphs, Architecture 1 is optimal according to this definition on a very general statistical model. We then compute the generalization error and compare the architecture to other well-studied methods like a simple MLP and a GCN.

### 3.1 Asymptotic Local Bayes Optimality

For classification tasks, it is natural to use a notion of generalization error in a "per sample" or online sense. Without graphical side information, the natural choice is the Bayes risk. With graphical information, however, there is an important obstruction: the number of samples is equal to the size of the corresponding graph. As such, a naive extension of the Bayes risk does not have this property.

A natural approach would be to consider the Bayes risk for estimators that take in the node, the data set, and the graph, i.e., $\hat{y}_v = \hat{y}(v, (X, G))$. In this case, however, the risk necessarily implicitly depends on the sample size, $n$, through $G$. One might try to remove this dependence by taking the infinite sample size limit, but for a class of estimators this general, it is not clear that such a limit is well defined. To circumvent this issue, we restrict attention to node classifiers that are only allowed "local" information around the node. The large graph limit of the generalization error is then naturally interpreted via *local weak convergence*. (For the convenience of the reader, we briefly recall the notion of local weak convergence of sparse graphs in Appendix A.1. See also Ramanan [2021, Chapter 1] or Bordenave [2016, Section 3] for more detailed expositions.) In this limit, one can then interpret the generalization error as a per-sample error for the randomly rooted graph $(G, u)$ where $u$ is a uniform random vertex in $V(G)$. (Here and in the following a *rooted graph* is a pair of a graph $G$

and a distinguished vertex, $u$, called *the root*.) With these observations, we are led to a natural notion of Bayes optimality, namely *asymptotic local Bayes optimality* which we define presently.[1]

Before turning to this definition, we must first recall the notion of $\ell$-*local classifiers*. For a node $v$ in a graph $G$, let $\eta_k(v) = \{u \in V(G) : d(u,v) \leq k\}$ denote the ball of radius $k$ for the canonical graph distance metric.

**Definition 3.1** ($\ell$-local classifier)**.** Let $G = (\mathbf{A}, \mathbf{X})$ be a feature-decorated graph of $n$ vertices with $d$-dimensional features $\mathbf{X}_u$ for each vertex $u$. For a fixed radius $\ell > 0$, an $\ell$-local node-classifier is a function $f$ that takes as input a root vertex $u \in [n]$, the adjacency matrix $\mathbf{A}$ and the features of all nodes within the $\ell$-neighbourhood of $u$, i.e., $\{\mathbf{X}_v\}_{v \in \eta_\ell(u)}$), and outputs a classification label for $u$.

Suppose now that we have a sequence of (random) feature decorated graphs $(X_n, G_n)$ with $|V(G)| = n$. Let $u_n$ denote a uniform at random vertex in $G_n$. Suppose finally that the rooted feature-decorated graphs $(X_n, G_n, u_n)$ locally weakly converge to $(X, G, u)$. We can then define the notion of asymptotically $\ell$-locally Bayes optimal classifiers for this sequence of problems.

**Definition 3.2.** We say that a classifier $h_\ell^* \in \mathcal{C}_\ell$ is asymptotically $\ell$-locally Bayes optimal classifier of the root for the sequence $\{(X_n, G_n, u_n)\}$ if it minimizes the probability of misclassification of the root of the local weak limit, $(X, G, u)$, over the class $\mathcal{C}_\ell$, i.e.,

$$h_\ell^* = \operatorname*{argmin}_{h \in \mathcal{C}_\ell} \mathbf{Pr}\left[ h(u, \{\mathbf{X}_v\}_{v \in \eta_\ell(u,G)}) \neq y_u \right].$$

Before turning to our data model, we note here that the reader may ask whether or not the asymptotically $\ell$-locally Bayes optimal classifier is in any sense the limit of optimal $\ell$-local classifier of the random root, $u_n$. We show this in an appropriate sense in Theorem 4.

## 3.2 Data Model

Let us now turn to the data model that we use for our analysis. We work with the general multi-class contextual stochastic block model (CSBM) where each node belongs to one of $C$ different classes labelled $1, \ldots, C$, and the node features have arbitrary continuous or discrete distributions. This model with $C = 2$, along with a specialization to Gaussian features has been extensively studied in several works on (semi)-supervised node classification and unsupervised community detection, see, for example, Deshpande et al. [2018], Lu and Sen [2020], Baranwal et al. [2021], Wei et al. [2022], Fountoulakis et al. [2022b], Baranwal et al. [2023]. Informally, a CSBM consists of a coupling of a stochastic block model (SBM) [Holland et al., 1983] with a mixture model where the components of the mixture have arbitrary distributions and are associated with the blocks of the SBM.

More formally, let $n, d$ be positive integers such that $n$ denotes the number of nodes and $d$ denotes the dimension of the node features. Define $y_1, \ldots, y_n \in \{1, \ldots, C\}$ as the latent variables (class labels) to be inferred. We will assume that the latent variables have a uniform prior, i.e., $y_u \sim \text{Unif}(\{[C]\})$ for all $u$. For the relational part of the data, we have an undirected unweighted graph of $n$ nodes, $G = (V, E)$ with adjacency matrix $\mathbf{A} = (a_{uv})_{u,v \in [n]} \sim \text{SBM}(n, \mathbf{Q})$, where $\mathbf{Q} = \{q_{ij}\} \in [0,1]^{C \times C}$ is the edge-probability matrix, meaning that

$$\mathbf{Pr}(a_{uv} = 1 \mid y_u = i, y_v = j) = q_{ij}.$$

The node attributes, $\mathbf{X} \in \mathbb{R}^{n \times d}$ are sampled from a mixture of $C$ arbitrary continuous or discrete distributions, $\mathbb{P} = \{\mathbb{P}_i\}_{i \in [C]}$, where corresponding to the $y_u$, we have $\mathbf{X}_u \sim \mathbb{P}_{y_u}$ for all $u \in [n]$.

We will view $n$ as large and study the setting where $d$ is fixed (does not grow with $n$). We note here that in previous related works [Baranwal et al., 2021, Wei et al., 2022, Baranwal et al., 2023], crucial assumptions have been made about the distribution of the node features and the sparsity of the graph, i.e., $q_{ij} = \Omega_n(\log^2 n/n)$. In contrast, we work in the extremely sparse setting where $q_{ij} = b_{ij}/n$ for constants $b_{ij} > 1$, so we write $\mathbf{Q} = \mathbf{B}/n$ where $\mathbf{B} = \{b_{ij}\}_{i,j \in [C]}$. Furthermore, the only assumption we need about the distributions $\mathbb{P}_i$ is that $\mathbb{P}_i$ are absolutely continuous with respect to some base measure, in which case their densities exist, denoted by $\rho_i$. For ease of reading, we encourage the

---

[1]It is also desirable for the empirical misclassification error to converge to the generalization error. If one works with the stronger notion of local convergence in probability, then this will hold as well. This later mode will hold in our examples but we leave this to future work.

reader to consider the case where $\mathbb{P}_i$ are continuous or discrete, therefore, the base measure is simply the Lebesgue measure on $\mathbb{R}$ or the counting measure on $\mathbb{Z}$ respectively.

For a feature-decorated graph $G = (\mathbf{A}, \mathbf{X}) = (\{a_{uv}\}_{u,v \in [n]}, \{\mathbf{X}_u\}_{u \in n})$ sampled from the model described above, we say that $G \sim \mathrm{CSBM}(n, d, \mathbb{P}, \mathbf{Q})$ or $G \sim \mathrm{CSBM}(n, d, \mathbb{P}, \mathbf{B}/n)$.

### 3.3 Optimal Classifier

We are now ready to state our first main result that characterizes the asymptotically $\ell$-locally Bayes optimal classifier on the CSBM data described in Section 3.2.

**Theorem 1** (Bayes optimal message-passing). *For any $\ell \geq 1$, the asymptotically $\ell$-locally Bayes optimal classifier of the root for the sequence $(G_n, u_n) \sim \mathrm{CSBM}(n, d, \mathbb{P}, \mathbf{Q})$ is*

$$h_\ell^*(u, \{\mathbf{X}_v\}_{v \in \eta_\ell(u)}) = \underset{i \in [C]}{\mathrm{argmax}} \Big\{ \log \rho_i(\mathbf{X}_u) + \sum_{v \in \eta_\ell(u) \setminus \{u\}} \mathcal{M}_{i \, d(u,v)}(\mathbf{X}_v) \Big\},$$

*where $\{\rho_i\}_{i \in [C]}$ are the densities associated with the distributions $\mathbb{P}_i \in \mathbb{P}$, and*

$$\mathcal{M}_{ik}(\mathbf{x}) = \max_{j \in [C]} \big\{ \log \rho_j(\mathbf{x}) + \log \mathbf{Q}_{ij}^k \big\}.$$

Let us briefly discuss the meaning of Theorem 1. It states that universally among all $\ell$-local classifiers, $h_\ell^*$ is asymptotically Bayes optimal for the sparse CSBM data. We view $\mathcal{M}_{ik}(\mathbf{X}_v)$ as the message gathered from node $v$ that is distance $k$ away from node $u$. In particular, $\mathcal{M}_{ik}(\mathbf{X}_v)$ naturally maximizes the likelihood of observing node $v$ in class $j$ at distance $k$ from node $u$ in class $i$, over all $j \in [C]$. Furthermore, this optimal classifier is realizable using Architecture 1 (see for example, Lu et al. [2017, Theorem 1], where it is shown that any Lebesgue measurable function can be approximated arbitrarily closely by standard neural networks). Consequently, this result shows that in the sparse setting, the message-passing paradigm can realize the optimal node classification scheme irrespective of the distributions of the node features or the inter-class edge probabilities.

For an intuitive understanding of Theorem 1, it helps to consider two extreme cases. First, if $\mathbf{Q} = p\mathbf{I}$ for some $p \in [0, 1]$, then the classifier reduces to a simple convolution, $h_\ell^*(u) = \mathrm{argmax}_{i \in [C]} \big\{ \sum_{v \in \eta_k(u)} \log \rho_i(\mathbf{X}_v) \big\}$. Second, if $\mathbf{Q} = p\mathbf{1}\mathbf{1}^\top$, then $q_{ij} = p$ for all $i, j \in [C]$, meaning that the graph component of the data is Erdös-Rényi, and hence, completely uninformative for the purposes of node classification. In this case, the classifier reduces to $h_\ell^*(u) = \mathrm{argmax}_{i \in [C]} \{ \log \rho_i(\mathbf{X}_u) \}$, i.e., it is optimal to look at only the features of node $u$ to predict its label since the neighbourhood does not provide any meaningful information. We formalize this intuition later for a simpler case (see Theorem 3).

### 3.4 Comparative Study

In this section, we perform a theoretical analysis of the classifier in Theorem 1 using a well-studied specialization of the CSBM data model described in Section 3.2. For ease of discussion, let us restrict ourselves to the setting where there are two classes. Formally, we have $C = 2$, and without loss of generality, the class labels $y_u \in \{\pm 1\}$ for all $u \in [n]$. The distributions of the node features are given by $\mathbf{X}_u \sim \mathbb{P}_{y_u}$ with corresponding density $\rho_{y_u}$. Furthermore, $\mathbf{Q} = \{q_{ij}\}$ is a $2 \times 2$ matrix with $q_{ii} = p = a/n$ and $q_{ij} = q = b/n$ with constants $a > 1, b \geq 0$ for classes $i \neq j$. For a data sample $G = (\mathbf{A}, \mathbf{X})$ from this model, we write $G \sim \mathrm{CSBM}(n, d, \{\mathbb{P}_\pm\}, \mathbf{Q})$ or $G \sim \mathrm{CSBM}(n, d, \{\mathbb{P}_\pm\}, \frac{a}{n}, \frac{b}{n})$. We also recognize the quantity associated with the signal-to-noise ratio (SNR) in the graph structure for this case, which is given by

$$\Gamma = \frac{|p - q|}{p + q} = \frac{|a - b|}{a + b}. \tag{1}$$

Note that the quantity $\Gamma$ has been recognized as the meaningful SNR in several related works where the underlying random graph model is the binary symmetric stochastic block model, for example, Baranwal et al. [2021], Fountoulakis et al. [2022b], Wei et al. [2022], Baranwal et al. [2023].

Let us now state Theorem 1 in the case of two classes. For given input $x \in \mathbb{R}$ and $c > 0$, let $\varphi(x, c) = \min(\max(x, -c), c)$ denote the value of $x$ clipped between the range $[-c, c]$.

**Corollary 1.1** (Optimal classifier for binary symmetric CSBM). *For any $\ell \geq 1$, the asymptotically $\ell$-locally Bayes optimal classifier of the root for the sequence $(G_n, u_n) \sim \text{CSBM}(n, d, \mathbb{P}, \frac{a}{n}, \frac{b}{n})$ is*

$$h_\ell^*(u, \{\mathbf{X}_v\}_{v \in \eta_\ell(u)}) = \text{sgn}\left(\psi(\mathbf{X}_u) + \sum_{v \in \eta_\ell(u) \setminus \{u\}} \mathcal{M}_{d(u,v)}(\mathbf{X}_v)\right),$$

*where $\mathcal{M}_k(\mathbf{x}) = \text{sgn}(a - b) \cdot \varphi(\psi(\mathbf{x}), c_k)$ with $c_k = \log\left(\frac{1 + \Gamma^k}{1 - \Gamma^k}\right)$, and $\psi(\mathbf{x}) = \log \frac{\rho_+(\mathbf{x})}{\rho_-(\mathbf{x})}$.*

In this simplified setting, we note that the messages propagated from nodes in the $\ell$-local neighbourhood of node $u$ are clipped proportional to a function of their distance $k$ from node $u$. In particular, the clip threshold $c_k$ can be expressed in terms of the graph SNR $\Gamma$ from (1). It is interesting to observe in Corollary 1.1 that $c_k$ decreases rapidly as $k$ increases. Since $\Gamma < 1$, this means that to predict the label of node $u$, the value of the message propagated from node $v$ at distance $k$ from $u$ decreases exponentially in $k$.

The above simplification helps us interpret the classifier in terms of the graph SNR $\Gamma$. We will now impose an assumption on the distribution of node features. This will help us analyze the generalization error in terms of the SNR in both the features and the graph, and enable us to compare the performance with other learning methods that are well-studied in the same statistical settings. We will resort to the setting where the features of the CSBM follow a Gaussian mixture. Note that this specialized statistical model has been studied extensively in previous works for benchmarking existing GNN architectures, see for example, Baranwal et al. [2021], Fountoulakis et al. [2022b], Wei et al. [2022], Baranwal et al. [2023].

In principle, one could compute the generalization error of $h_\ell^*$ for arbitrary distributions on the node features (see Appendix A.3.2), however, we report the error for Gaussian features for expository reasons. The generalization error is defined for a classifier $h$ to be the probability of disagreement between the true label $y_u$ and the output of the classifier $h_u$ for node $u$. We characterize the error for $h_\ell^*$ in the case where $\mathbb{P}_-, \mathbb{P}_+$ correspond to the Gaussian mixture with components $\mathcal{N}(-\boldsymbol{\mu}, \sigma^2 \mathbf{I})$ and $\mathcal{N}(\boldsymbol{\mu}, \sigma^2 \mathbf{I})$ for fixed $\boldsymbol{\mu} \in \mathbb{R}^d$ and $\sigma > 0$[2].

In this case, a notion of the signal-to-noise ratio of the features naturally exists, i.e., $\gamma = \|\boldsymbol{\mu}\|_2 / \sigma$, a quantity proportional to the ratio of the distance between the means of the mixture and the standard deviation. The log-likelihood ratio in this setting is $\psi(\mathbf{x}) = \log \frac{\rho_+(\mathbf{x})}{\rho_-(\mathbf{x})} = \frac{2}{\sigma^2} \langle \mathbf{x}, \boldsymbol{\mu} \rangle$.

Consider a sequence $\{(G_n, u_n)\}_{n \geq 1}$ with $G_n = (V(G_n), E(G_n))$ from this model where $u_n \sim \text{Unif}(V(G_n))$. In this setting, in the absence of features, it is known that $(G_n, u_n)$ converges locally weakly to a Poisson Galton-Watson tree (see for example, Mossel et al. [2015, Section 4]). Here, for every node, we additionally have features that are independent of the graph, and hence, as a straightforward consequence of Mossel et al. [2015, Section 4], $(G_n, u_n)$ in our case converges to a feature-decorated Poisson Galton-Watson tree $(G, u)$.

For the root node $u$, let $\alpha_k$ and $\beta_k$ denote the number of children at generation $k$ in class $y_u$ and $-y_u$ respectively, where $y_u$ denotes the label of node $u$. Then $\{\alpha_k\}_{k \geq 0}$ and $\{\beta_k\}_{k \geq 0}$ are characterized by

$$\alpha_0 = 1, \beta_0 = 0,$$
$$\alpha_k \sim \text{Poi}\left(\frac{a\alpha_{k-1} + b\beta_{k-1}}{2}\right), \beta_k \sim \text{Poi}\left(\frac{a\beta_{k-1} + b\alpha_{k-1}}{2}\right) \text{ for } k \in [\ell]. \tag{2}$$

For a classifier $h$ acting on $G$, let $\mathcal{E}(h)$ denote the probability of misclassification of the root $u$ in $G$, i.e., $\mathcal{E}(h) = \mathbf{Pr}(h_u y_u < 0)$. Correspondingly, in the case of finite $n$, we denote by $\mathcal{E}_n(h)$ the probability of misclassification of a uniform random node $u_n$ in $G_n$. We are now ready to state the generalization error of $h_\ell^*$.

**Theorem 2** (Generalization error). *For any $\ell \geq 1$, the generalization error of the asymptotically $\ell$-locally Bayes optimal classifier of the root for the sequence $(G_n, u_n) \sim \text{CSBM}(n, d, \mathbb{P}, \mathbf{Q})$ with Gaussian features is given by*

$$\mathcal{E}(h_\ell^*) = \mathbf{Pr}\left[g + \frac{1}{2\gamma} \sum_{k \in [\ell]} \left(\sum_{i \in [\alpha_k]} Z_{k,i}^{(a)} + \sum_{i \in [\beta_k]} Z_{k,i}^{(b)}\right) > \gamma\right],$$

---

[2]In principle, it is easy to analyze arbitrarily fixed means, say $\boldsymbol{\mu}, \boldsymbol{\nu}$ instead of keeping them symmetric, i.e., $\pm\boldsymbol{\mu}$, with only minor changes in calculations.

where $\alpha_k, \beta_k$ are as in (2), $Z_{k,i}^{(a)} = \varphi(-2\gamma^2 + 2\gamma g_{k,i}, c_k)$, $Z_{k,i}^{(b)} = \varphi(2\gamma^2 + 2\gamma g_{k,i}, c_k)$, and $g, \{g_{k,i}\}$ are mutually independent standard Gaussian random variables.

Let us now understand how the error described in Theorem 2 behaves in terms of the two SNRs $\gamma$ (for the features), and $\Gamma$ (for the graph). Note that $\mathcal{E}(h_\ell^*) \to 0$ as $\gamma \to \infty$, and $\mathcal{E}(h_\ell^*) \to 1/2$ as $\gamma \to 0$. This means that if the signal in the features is large, the number of mistakes made by the classifier vanishes, while if the signal is very small, then roughly half of the nodes are misclassified (equivalent to making a uniform random guess for each node).

To see how $\Gamma$ affects the error, we begin by looking at two extreme settings: first, where the graph is complete noise, i.e., $\Gamma = 0$, and second, where the graph signal is very strong, i.e., $\Gamma \to 1$, followed by a discussion on how $h_\ell^*$ interpolates between these extremes. Let $\phi_\pm$ denote the Gaussian density functions with means $\pm\boldsymbol{\mu}$ and variance $\sigma^2 \mathbf{I}_d$. Define the random variable

$$\xi_\ell = \xi_\ell(a, b) = \frac{1 + \sum_{k=1}^\ell |\alpha_k - \beta_k|}{\sqrt{1 + \sum_{k=1}^\ell (\alpha_k + \beta_k)}}, \tag{3}$$

where $\alpha_k, \beta_k$ follow (2). In the following, we denote the vanilla GCN classifier from Kipf and Welling [2017] by $h_{\text{gcn}}$. We then have the following result.

**Theorem 3** (Extreme graph signals). *Let $h_\ell^*$ be the classifier from Corollary 1.1, $h_0^*(u) = \text{sgn}(\langle \mathbf{X}_u, \boldsymbol{\mu} \rangle)$ be the Bayes optimal classifier given only the feature information of the root node $u$, and $h_{\text{gcn}}$ be the one-layer vanilla GCN classifier. Then we have that for any fixed $\ell$:*

1. *If $\Gamma = 0$ then $\mathcal{E}(h_\ell^*) = \mathcal{E}(h_0^*) = \Phi(-\gamma)$, where $\Phi$ is the standard Gaussian CDF.*

2. *If $\Gamma \to 1$ then $\xi_\ell \geq 1$ a.s. and $\mathcal{E}(h_\ell^*) \to \mathbf{Pr}(g > \gamma\xi_\ell)$, where $g \sim \mathcal{N}(0, 1)$.*

3. *$\mathcal{E}(h_{\text{gcn}}) = \mathbf{Pr}(g > \gamma\xi_1)$.*

Theorem 3 shows that in the regime of extremely low graph SNR, the optimal classifier $h_\ell^*$ reduces to a linear classifier $h_0^*(u) = \text{sgn}(\langle \mathbf{X}_u, \boldsymbol{\mu} \rangle)$, which can be realized by a simple MLP that does not use the graph component of the data at all. On the other hand, in the regime of extremely strong graph SNR, $h_\ell^*$ reduces to a simple convolution over all nodes in the $\ell$-neighbourhood and is comparable to a typical GCN. Furthermore, we note that in the strong graph SNR regime $\Gamma \to 1$, $\mathcal{E}(h_\ell^*) \to \mathbf{Pr}(g > \gamma\xi_\ell) \leq \Phi(-\gamma)$ since $\xi_\ell \geq 1$. The clip operation during the propagation of messages makes things interesting between these two extremes, where $h_\ell^*$ interpolates between a simple MLP and an $\ell$-hop convolutional network. This interpolation is characterized by the graph signal $\Gamma$, since the messages are clipped in the range $[-c_k, c_k]$, where $c_k = \log\left(\frac{1+\Gamma^k}{1-\Gamma^k}\right)$.

In addition, Theorem 3 concludes that if $\xi_1(a, b) > 1$, then a GCN can perform better than every classifier that does not see the graph. On the other hand, if $\xi_1(a, b) < 1$, then a GCN incurs more errors on the data than the best methods that do not use the graph. Interestingly, but not surprisingly, this result aligns with the Kesten-Stigum weak recovery threshold for the community-detection problem on the sparse stochastic block model [Massoulié, 2014, Mossel et al., 2018], meaning that if weak recovery is possible on the graph component of the data, then a GCN is able to exploit it to perform better than methods that do not use the graph, e.g., a simple MLP.

We now demonstrate our results through experiments using `pytorch` and `pytorch-geometric` [Fey and Lenssen, 2019]. The following simulations are for the setting $n = 10000$ and $d = 4$ for binary classification on the CSBM. We implement Architecture 1 for the binary case, and perform full-batch training on a graph sampled from the CSBM with certain signals (mentioned in the figures), followed by an evaluation of the architecture on a new graph sampled from the same distribution.

In Fig. 1, we show that the accuracy obtained by the optimal classifier is higher than both a simple MLP and a vanilla GCN [Kipf and Welling, 2017]. We plot the test accuracy of Architecture 1 against the SNR in the node features, $\gamma = \|\boldsymbol{\mu}\| / \sigma$ in Fig. 1a, and against the graph SNR $\Gamma = |a - b|/(a + b)$ in Fig. 1b. We fix $\Gamma = 0.42$ and $\gamma = 1$ for the two plots, respectively. We chose these specific values because they generate relatively clearer plots where the accuracy metrics for the three architectures are easily visible and distinguished from each other. The results are similar for other values for $\Gamma$ and $\gamma$, i.e., the Bayes optimal architecture is superior to both MLP and GCN.

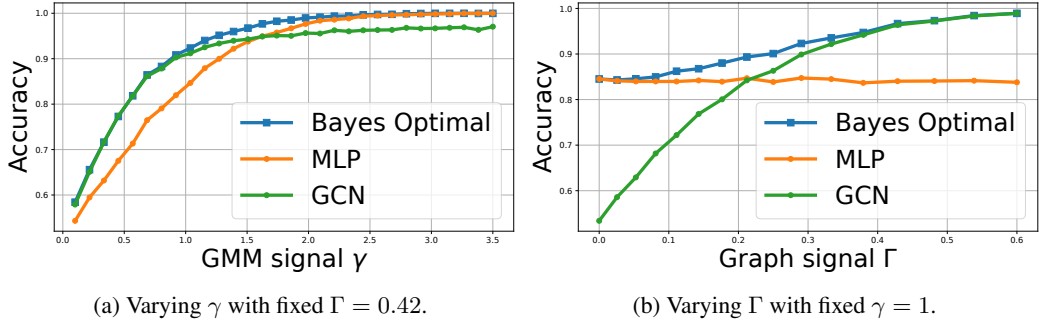

(a) Varying $\gamma$ with fixed $\Gamma = 0.42$.

(b) Varying $\Gamma$ with fixed $\gamma = 1$.

Figure 1: Comparison of Architecture 1 against an MLP and a vanilla GCN [Kipf and Welling, 2017].

Furthermore, Fig. 2 shows that as claimed in Theorem 3, when the graph signal is at the extremes, i.e., $\Gamma = 0$ and $\Gamma = 1$, Architecture 1 behaves like a simple MLP and performs a typical convolution (averaging) over all nodes in the $\ell$-neighbourhood, respectively. In the regime of poor graph SNR, i.e., $\Gamma = 0$, a GCN is worse than a simple MLP, as inferred from part three of Theorem 3.

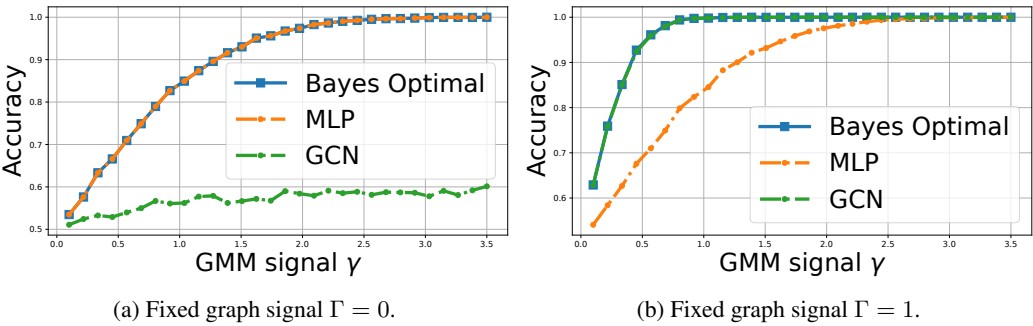

(a) Fixed graph signal $\Gamma = 0$.

(b) Fixed graph signal $\Gamma = 1$.

Figure 2: Demonstration of Theorem 3 for extreme graph signals. In the case where $\Gamma = 0$, the architecture reduces to an MLP (Fig. 2a), while if $\Gamma = 1$, it behaves the same as a GCN (Fig. 2b).

Finally, we observe that for the binary setting when the parameters of the architecture are initialized uniformly at random, gradient descent converges and the neural network learns the right parameters such that Architecture 1 realizes the optimal classifier in Corollary 1.1. This convergence, along with a comparison of our architecture to the Approximate Message-passing Belief Propagation (AMP-BP) algorithm from Deshpande et al. [2018] is presented in Appendix B.

## 3.5  Non-asymptotic Setting

We now turn to the non-asymptotic regime and argue that for fixed $n$, the classifier in Corollary 1.1 is still in a formal sense, Bayes optimal for an overwhelming fraction of nodes. We begin by exploiting the fact that for up to logarithmic depth neighbourhoods, a sparse CSBM graph is tree-like.

**Proposition 3.1** (Tree neighbourhoods)**.** *Let $G = (V, E) \sim \text{CSBM}(n, d, \mathbb{P}, \frac{a}{n}, \frac{b}{n})$ for constants $a, b > 1$. Then for any $\ell = c \log n$ such that $c \log((a+b)/2) < 1/4$, with probability $1 - O(1/\log^2 n)$, the number of nodes $u \in V$ whose $\ell$-neighbourhood is cycle-free is $n(1 - o(\log^4(n)/\sqrt{n}))$.*

In particular, Proposition 3.1 states that for $\ell = c \log n$ for a suitable constant $c$, the $\ell$-neighbourhood of an overwhelming fraction of nodes is a tree. This implies that the classifier $h_\ell^*$ is Bayes optimal for roughly all of the nodes. Moreover, since the diameter of a sparse graph (as in our setting) is $O(\log n)$ almost surely [Chung and Lu, 2001, Theorem 6], any learning mechanism can only look as far as $O(\log n)$-hops away from a node to gather new information. This shows that for such graphs, GNNs that are not very deep and look at only up to logarithmic distance in the neighbourhood are sufficient.

Let us now turn to the misclassification error in the non-asymptotic setting. Recall that for a classifier $h \in \mathcal{C}_\ell$, we denote by $\mathcal{E}_n(h)$ and $\mathcal{E}(h)$ the misclassification error of $h$ on the data model with $n$ nodes,

and on the limiting data model with $n \to \infty$, respectively. Furthermore, recall from Corollary 1.1 that $\min_{h \in \mathcal{C}_\ell} \mathcal{E}(h) = \mathcal{E}(h_\ell^*)$. Our next result shows that the optimal misclassification error in the non-asymptotic setting across all $\ell$-local classifiers, i.e., $\min_{h \in \mathcal{C}_\ell} \mathcal{E}_n(h)$, is close to the misclassification error obtained in the non-asymptotic setting by $h_\ell^*$. Moreover, $\min_{h \in \mathcal{C}_\ell} \mathcal{E}_n(h)$ is also close to $\mathcal{E}(h_\ell^*)$ which is explicitly computed in Theorem 2.

**Theorem 4** (Misclassification error for fixed $n$). *For any $1 \leq \ell \leq c \log n$ such that the positive constant $c$ satisfies $c \log(\frac{a+b}{2}) < 1/4$, we have that*

$$\left| \min_{h \in \mathcal{C}_\ell} \mathcal{E}_n(h) - \mathcal{E}_n(h_\ell^*) \right| = O\left( \frac{1}{\log^2 n} \right), \quad \left| \min_{h \in \mathcal{C}_\ell} \mathcal{E}_n(h) - \mathcal{E}(h_\ell^*) \right| = O\left( \frac{1}{\log^2 n} \right).$$

Recall that Corollary 1.1 implies that $h_\ell^*$ performs optimally on the limiting data model (asymptotic setting) among the class of $\ell$-local classifiers $\mathcal{C}_\ell$, but it may not be optimal for the non-asymptotic data model where we have a finite feature-decorated graph with $n$ nodes. However, Theorem 4 helps us conclude that even in the non-asymptotic setting, $h_\ell^*$ performs almost as well as the actual optimal classifier among $\mathcal{C}_\ell$ in this case, as long as we compare with classifiers that can only look at moderate logarithmic depths in the local neighbourhood, i.e., $\ell \leq c \log n$ for a suitable $c$.

# 4 Conclusion and Future Work

In this work, we present a comprehensive theoretical characterization of the Bayes optimal node classification architecture for sparse feature-decorated graphs and show that it can be realized using the message-passing framework. Utilizing a well-established and well-studied statistical model, we interpret its performance in terms of the SNR in the data and validate our findings through empirical analysis of synthetic data. Additionally, we identify the following limitations as prospects for future work: (1) We consider neighbourhoods up to distance $\ell = c \log n$ for a small enough $c$. Extending $\ell$ to the graph's diameter (known to be $O(\log n)$ with high probability) by removing the restriction on $c$ poses challenges due to the presence of cycles. (2) More insights can be provided through experiments on real data to benchmark the architecture in cases where we have a significant gap between the theoretical assumptions (sparse and locally tree-like graph) and the real-world data.

## Acknowledgments and Disclosure of Funding

A.J. acknowledges the support of the Natural Sciences and Engineering Research Council of Canada (NSERC) and the Canada Research Chairs programme. Cette recherche a été enterprise grâce, en partie, au soutien financier du Conseil de Recherches en Sciences Naturelles et en Génie du Canada (CRSNG), [RGPIN-2020-04597, DGECR-2020-00199], et du Programme des chaires de recherche du Canada.

K. Fountoulakis would like to acknowledge the support of the Natural Sciences and Engineering Research Council of Canada (NSERC). Cette recherche a été financée par le Conseil de recherches en sciences naturelles et en génie du Canada (CRSNG), [RGPIN-2019-04067, DGECR-2019-00147].

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

# A  Supplementary Discussion and Proofs

## A.1  Local Weak Convergence

We briefly recall here the notion of local weak convergence of random rooted graphs. The notion of local weak convergence of random, feature-decorated, rooted graphs is defined analogously.

Let us begin first with the case of rooted graphs. A rooted graph $(G, u) = ((E, V), u)$ is a graph $G$ with a distinguished vertex $u$ called the root. We say that two rooted graphs $(G_1, u_1) = ((E_1, V_1), u_1)$, $(G_2, u_2) = ((E_2, V_2), u_2)$ are isomorphic if there is a bijection $\phi : V_1 \to V_2$ such that $\phi(u) = v$ and such that if $(x, y) \in E_1$ then $(\phi(x), \phi(y)) \in E_2$. In this case we write $(G_1, u_1) \cong (G_2, u_2)$. For a rooted graph $(G, u)$, we denote its isomorphism class as $[(G, u)]$.

Let $\mathcal{G}_*$ denote the set of isomorphism classes of (locally finite) rooted graphs. For a vertex $v \in G$ we let $\eta_k(v)$ denote the collection of neighbours of $v$ of distance at most $k$ in the canonical edge distance metric, and $G[\eta_k(v)]$ denote the subgraph induced on this collection of vertices. We then have the notion of *local convergence* in $\mathcal{G}_*$.

**Definition A.1** (Local convergence on $\mathcal{G}_*$). We say that a sequence $[(G_n, u_n)] \in \mathcal{G}_*$ *converges locally* to $[(G, u)] \in \mathcal{G}_*$ if for each $k > 0$, we have that $G[\eta_k(u_n)] \cong G[\eta_k(u)]$ eventually.

It can be shown [Bordenave, 2016, Lemma 3.4] that $\mathcal{G}_*$ equipped with the topology of local convergence is a Polish space. We are then in the position to define local weak convergence on $\mathcal{G}_*$. In brief, the topology of local weak convergence of random graphs is the topology of weak convergence of measures on the space of probability measures on $\mathcal{G}_*$, namely $\mathcal{M}_1(\mathcal{G}_*)$.

**Definition A.2** (local weak convergence of rooted graphs). A sequence $\{[G_n, u_n]\}$ of random rooted graphs with corresponding laws $\{\mu_n\} \subseteq \mathcal{M}_1(\mathcal{G}_*)$ is said to locally weakly converge to a random rooted graph $[G, u]$ with law $\mu \in \mathcal{M}_1(\mathcal{G}_*)$ if $\mu_n \to \mu$ weakly.

We note here that it is common also to talk about the notion of local weak convergence of a sequence of (finite) random graphs $G_n$. In this case $G_n \to G$ locally weakly if $[(G_n, u_n)] \to [(G, u)]$ locally weakly where $u_n \sim \mathrm{Unif}(V(G_n))$.

## A.2  Preliminary Results

In this section, we state two important preliminary results and a fact that are used to establish our results in the paper.

**Lemma A.1.** *[Dreier et al., 2018, Lemma 3]. Let $G \sim \mathrm{CSBM}(n, d, \{\mathbb{P}_-, \mathbb{P}_+\}, \frac{a}{n}, \frac{b}{n})$ and $r$ be a fixed constant. Then the probability that there exists an $r$-neighbourhood in $G$ with $m$ more edges than the number of vertices is bounded as follows:*

$$\mathbf{Pr}(\exists\, G_r \subset G \text{ s.t. } |E(G_r)| \geq |V(G_r)| + m) \leq \frac{(2(r(m+1))^2(a+b))^{2r(m+1)+m}}{n^m}.$$

**Lemma A.2.** *[Massoulié, 2014, Lemma 4.2]. Assume $\ell = c\log(n)$ with $c\log\Delta < 1/4$, where $\Delta = (a+b)/2$. Then with high probability, no node $i$ has more than one cycle in its $\ell$-neighbourhood. Moreover, for any $m > 0$, with probability at least $1 - O(1/m^2)$ the number of nodes $i$ whose $\ell$-neighbourhood contains at least one cycle is bounded by $O(m\log^3(n)\Delta^{2\ell})$.*

**Fact A.3.** *For any non-negative $u, v, x, y$ such that $x, y \leq 1$,*

$$xu + yv - \frac{1}{2}(xu + yv)^2 \leq 1 - (1-x)^u(1-y)^v \leq xu + yv.$$

## A.3  Bayes Optimal Classifier

In this section, we compute the asymptotically $\ell$-locally Bayes optimal classifier for the general CSBM described in Section 3.2 and establish Theorem 1, followed by a proof of Corollary 1.1. Next, we compute the generalization error for the two-class case with arbitrary node features.

### A.3.1  Computing the Classifier

For the proofs, we introduce the notation $N_k(u)$ for a given graph to mean the set of vertices that are at a distance of exactly $k$ from node $u$ in the graph. Thus, $\eta_k(u) = \{u\} \cup_{j=1}^k N_k(u)$.

**Theorem** (Restatement of Theorem 1). *For any $\ell \geq 1$, the asymptotically $\ell$-locally Bayes optimal classifier of the root for the sequence $(G_n, u_n) \sim \mathrm{CSBM}(n, d, \mathbb{P}, \mathbf{Q})$ is*

$$h_\ell^*(u, \{\mathbf{X}_v\}_{v \in \eta_\ell(u)}) = \underset{i \in [C]}{\mathrm{argmax}} \Big\{ \log \rho_i(\mathbf{X}_u) + \sum_{v \in \eta_\ell(u) \setminus \{u\}} \mathcal{M}_{i\,d(u,v)}(\mathbf{X}_v) \Big\},$$

*where $\{\rho_i\}_{i \in [C]}$ are the densities associated with the distributions $\mathbb{P}_i \in \mathbb{P}$, and*

$$\mathcal{M}_{ik}(\mathbf{x}) = \max_{j \in [C]} \left\{ \log \rho_j(\mathbf{x}) + \log \mathbf{Q}_{ij}^k \right\}.$$

*Furthermore, there exists a choice of parameters for Architecture 1 such that it realizes $h_\ell^*$.*

*Proof.* We begin by writing the MAP estimation for this problem. Note that the features $\mathbf{X}_v \in \mathbb{R}^d$ for every node $v \in [n]$ follow the law $\mathbb{P}_i$ if $y_v = i \in [C]$. In addition, recall that we have the edge-probability matrix $\mathbf{Q} = \{q_{ij}\}_{i,j \in [C]}$ with $\mathbf{Pr}((u,v) \text{ is an edge} \mid y_u = i, y_v = j) = q_{ij}$, where $q_{ij} = b_{ij}/n$ for absolute constants $b_{ij}$. Then we can write the likelihood of the $\ell$-neighbourhood of node $u$ as the joint function:

$$f(u, \{(\mathbf{X}_v, y_v)\}_{v \in \eta_\ell(u)}, \eta_\ell(u)) = \mathbf{P}_{\{y_v\}_{v \in \eta_\ell(u)}} \rho_{y_u}(\mathbf{X}_u) \prod_{k \in [\ell]} \prod_{v \in N_k(u)} \rho_{y_v}(\mathbf{X}_v) \mathbf{Q}_{y_u y_v}^k. \qquad (4)$$

In the above, $\mathbf{Q}_{ij}^k$ denotes the $i,j$-th entry of the matrix $\mathbf{Q}^k$ and quantifies the probability that a node in class $j$ is at a distance $k$ from a node in class $i$; while $\mathbf{P}_{\{y_v\}_{v \in \eta_\ell(u)}}$ denotes the prior distribution of the node labels, which by our assumption is uniform. Let us now compute the MAP estimator.

$$
\begin{aligned}
h_\ell^*(u, \{\mathbf{X}_v\}_{v \in \eta_\ell(u)}) &= \underset{i \in [C]}{\mathrm{argmax}} \max_{\substack{\{y_v \in [C]\} \\ v \in \eta_\ell(u)}} f(u, \{(\mathbf{X}_v, y_v)\}_{v \in \eta_\ell(u)}, \eta_\ell(u)) \\
&= \underset{i \in [C]}{\mathrm{argmax}} \, \rho_i(\mathbf{X}_u) \prod_{k \in [\ell]} \prod_{v \in N_k(u)} \max_{j \in [C]} \rho_j(\mathbf{X}_v) \mathbf{Q}_{ij}^k \\
&= \underset{i \in [C]}{\mathrm{argmax}} \, \log \rho_i(\mathbf{X}_u) + \sum_{k \in [\ell]} \sum_{v \in N_k(u)} \max_{j \in [C]} \left\{ \log(\rho_j(\mathbf{X}_v)) + \log(\mathbf{Q}_{ij}^k) \right\} \\
&= \underset{i \in [C]}{\mathrm{argmax}} \, \log \rho_i(\mathbf{X}_u) + \sum_{k \in [\ell]} \sum_{v \in N_k(u)} \max_{j \in [C]} \mathcal{M}_{ik}(\mathbf{X}_v),
\end{aligned}
$$

where $\mathcal{M}_{ik}(\mathbf{X}_v) = \log(\rho_j(\mathbf{X}_v)) + \log(\mathbf{Q}_{ij}^k)$. Furthermore, note that an instance of Architecture 1 with $L = 1$, $\sigma_1 = \{\rho_i\}_{i \in [C]}$, and $\mathbf{Q}$ for the edge-probabilities realizes the function $h_\ell^*$ for a given root node $u$ and its $\ell$-neighbourhood $\eta_\ell(u)$, in the sense that $\hat{y}_u = h_\ell^*(u, \{\mathbf{X}_v\}_{v \in \eta_\ell(u)})$. $\qquad \square$

Next, we obtain a simpler version of the classifier for the two-class symmetric CSBM with an arbitrary distribution of node features. Recall that $\psi$ is the log of the likelihood ratio.

**Corollary** (Restatement of Corollary 1.1). *For any $\ell \geq 1$, the asymptotically $\ell$-locally Bayes optimal classifier of the root for the sequence $(G_n, u_n) \sim \mathrm{CSBM}(n, d, \mathbb{P}, \frac{a}{n}, \frac{b}{n})$ is*

$$h_\ell^*(u, \{\mathbf{X}_v\}_{v \in \eta_\ell(u)}) = \mathrm{sgn}\Big( \psi(\mathbf{X}_u) + \sum_{v \in \eta_\ell(u) \setminus \{u\}} \mathcal{M}_{d(u,v)}(\mathbf{X}_v) \Big),$$

*where $\mathcal{M}_k(\mathbf{x}) = \mathrm{sgn}(a - b) \cdot \varphi(\psi(\mathbf{x}), c_k)$ with $c_k = \log\left( \frac{1 + \Gamma^k}{1 - \Gamma^k} \right)$, and $\psi(\mathbf{x}) = \log \frac{\rho_+(\mathbf{x})}{\rho_-(\mathbf{x})}$.*

*Proof.* The proof follows directly from Theorem 1, by taking $C = 2$. In this two-class case, the features $\mathbf{X}_v \in \mathbb{R}^d$ for every node follow the law $\mathbb{P}_{y_v}$ where the class labels $y_v \in \{\mp 1\}$ are $\mathrm{Unif}(\{-1, +1\})$. In addition, we have

$$\mathbf{Pr}((u,v) \text{ is an edge}) = \begin{cases} p & y_u = y_v \\ q & y_u \neq y_v \end{cases},$$

where $p = a/n$ and $q = b/n$ for absolute constants $a > 1, b \geq 0$. Define the quantities $p_k, q_k$ as follows for $k \in [\ell]$:

$$p_k = \sum_{j=0}^{\lfloor k/2 \rfloor} \binom{k}{2j} p^{k-2j} q^{2j}, \qquad q_k = \sum_{j=1}^{\lceil k/2 \rceil} \binom{k}{2j-1} p^{k-2j+1} q^{2j-1}. \tag{5}$$

Then we can simplify the likelihood of the $\ell$-neighbourhood of node $u$ from (4) as follows:

$$f(\{(\mathbf{X}_v, y_v)\}_{v \in \eta_\ell(u)}) = \rho_{y_u}(\mathbf{X}_u) \prod_{k \in [\ell]} \prod_{v \in N_k(u)} \left( \rho_{y_v}(\mathbf{X}_v) p_k^{\frac{1+y_u y_v}{2}} q_k^{\frac{1-y_u y_v}{2}} \right).$$

Then maximizing the likelihood over possible class labels, we have

$$h_\ell^*(u, \{\mathbf{X}_v\}_{v \in \eta_\ell(u)}) = \underset{y_u \in \{\pm 1\}}{\operatorname{argmax}} \max_{\substack{\{y_v \in \{\pm 1\}\} \\ v \in \eta_\ell(u)}} f(\{(\mathbf{X}_v, y_v)\}_{v \in \eta_\ell(u)})$$

$$= \underset{y_u \in \{\pm 1\}}{\operatorname{argmax}} \max_{\substack{\{y_v \in \{\pm 1\}\} \\ v \in \eta_\ell(u)}} \log f(\{(\mathbf{X}_v, y_v)\}_{v \in \eta_\ell(u)})$$

$$= \underset{y_u \in \{\pm 1\}}{\operatorname{argmax}} \left\{ \log \rho_{y_u}(\mathbf{X}_u) + \max_{\substack{\{y_v \in \{\pm 1\}\} \\ v \in \eta_\ell(u)}} \sum_{k \in [\ell]} \sum_{v \in N_k(u)} \log \left( \rho_{y_v}(\mathbf{X}_v) p_k^{\frac{1+y_u y_v}{2}} q_k^{\frac{1-y_u y_v}{2}} \right) \right\}$$

$$= \operatorname{sgn} \left( \log \frac{\rho_+(\mathbf{X}_u)}{\rho_-(\mathbf{X}_u)} + \sum_{k \in [\ell]} \sum_{v \in N_k(u)} \mathcal{M}_k(\mathbf{X}_v) \right),$$

where $\mathcal{M}_k(\mathbf{x}) = \log(\max(p_k \rho_+(\mathbf{x}), q_k \rho_-(\mathbf{x}))) - \log(\max(p_k \rho_-(\mathbf{x}), q_k \rho_+(\mathbf{x})))$. Next, we observe that for any $w, x, y, z \in \mathbb{R}$,

$$\log(\max(wy, xz)) - \log(\max(wz, xy)) = \operatorname{sgn}(w - x) \cdot \min \left( \max \left( \log \frac{y}{z}, - \left| \log \frac{w}{x} \right| \right), \left| \log \frac{w}{x} \right| \right).$$

Hence, $\mathcal{M}_k(\mathbf{X}_v)$ is simply $\operatorname{sgn}(a - b) \cdot \varphi(\log \frac{\rho_+(\mathbf{X}_v)}{\rho_-(\mathbf{X}_v)}, c_k)$, i.e., the signed likelihood ratio clipped between $-c_k$ and $c_k$. □

### A.3.2 Generalization Error

Let us now compute the generalization error of $h_\ell^*$. Formally, given a data instance $(u, \{\mathbf{X}_v\}_{v \in \eta_\ell(u)})$ along with the neighbourhood $\eta_\ell(u)$, $h_\ell^*$ outputs a label $\hat{y}_u \in \{\pm 1\}$, and the generalization error is defined as the probability $\mathbf{Pr}(y_u \hat{y}_u < 1)$. For a simple calculation, let us assume that the latent labels $y_i$ are uniformly distributed, i.e., $\mathbf{Pr}(y_i = -1) = \mathbf{Pr}(y_i = 1) = \frac{1}{2}$. It is straightforward to generalize to unbalanced settings. Recall that the features in classes $\pm 1$ follow the law $\mathbb{P}_\pm$, and denote by the log likelihood ratio by $\psi(\mathbf{x}) = \log(\rho_+(\mathbf{x})/\rho_-(\mathbf{x}))$. Then we have that for a fixed $u$,

$$\mathcal{E}(h_\ell^*) = \mathbf{Pr} \left( y_u \left( \log \frac{\rho_+(\mathbf{X}_u)}{\rho_-(\mathbf{X}_u)} + \sum_{k \in [\ell]} \sum_{v \in N_k(u)} \mathcal{M}_k(\mathbf{X}_v) \right) < 0 \right)$$

$$= \frac{1}{2} \left[ \mathbf{Pr} \left( \psi(\mathbf{Y}^{(1)}) + \sum_{k \in [\ell]} \mathbf{Z}_k^{(1)} > 0 \right) + \mathbf{Pr} \left( \psi(\mathbf{Y}^{(2)}) + \sum_{k \in [\ell]} \mathbf{Z}_k^{(2)} < 0 \right) \right],$$

where $\mathbf{Z}_k^{(1)} = \sum_{j \in [\alpha_k]} \mathcal{M}_k(\mathbf{Y}_{k,j}^{(1)}) + \sum_{j \in [\beta_k]} \mathcal{M}_k(\mathbf{Y}_{k,j}^{(2)})$ and $\mathbf{Z}_k^{(2)} = \sum_{j \in [\alpha_k]} \mathcal{M}_k(\mathbf{Y}_{k,j}^{(2)}) + \sum_{j \in [\beta_k]} \mathcal{M}_k(\mathbf{Y}_{k,j}^{(1)})$ are independent random variables with $\mathbf{Y}_{k,j}^{(1)} \sim \mathbb{P}_-$ and $\mathbf{Y}_{k,j}^{(2)} \sim \mathbb{P}_+$.

The above expression is not particularly insightful. For this reason, we now specialize to the case of Gaussian features and interpret the error in terms of the natural SNRs associated with the Gaussian mixture and the graph, i.e., the quantities $\gamma$ and $\Gamma$.

### A.4 Specialization to Gaussian Features

In this section, we look at the specialized setting where the node features are sampled from a symmetric binary Gaussian mixture model. Let us begin with the generalization error in this case.

#### A.4.1 Generalization Error

**Theorem** (Restatement of Theorem 2). *For any $\ell \geq 1$, the generalization error of the asymptotically $\ell$-locally Bayes optimal classifier of the root for the sequence $(G_n, u_n) \sim \mathrm{CSBM}(n, d, \mathbb{P}, \mathbf{Q})$ with Gaussian features is given by*

$$\mathcal{E}(h_\ell^*) = \mathbf{Pr}\left[ g + \frac{1}{2\gamma} \sum_{k \in [\ell]} \left( \sum_{i \in [\alpha_k]} Z_{k,i}^{(a)} + \sum_{i \in [\beta_k]} Z_{k,i}^{(b)} \right) > \gamma \right],$$

*where $\alpha_k, \beta_k$ are as in (2), $Z_{k,i}^{(a)} = \varphi(-2\gamma^2 + 2\gamma g_{k,i}, c_k)$, $Z_{k,i}^{(b)} = \varphi(2\gamma^2 + 2\gamma g_{k,i}, c_k)$, and $g, \{g_{k,i}\}$ are mutually independent standard Gaussian random variables.*

*Proof.* For the Gaussian mixture, the log of the likelihood ratio for a node $u$ is given by $\frac{2}{\sigma^2}\langle \mathbf{X}_u, \boldsymbol{\mu} \rangle \stackrel{\mathcal{D}}{=} 2y_u\gamma^2 + 2\gamma g$, where $g \sim \mathcal{N}(0,1)$. Replacing every $\{\mathbf{X}_i\}_{i \in [n]}$ as $\mathbf{X}_i = \mathbb{E}\mathbf{X}_i + \sigma\mathbf{g}_i = y_i\boldsymbol{\mu} + \sigma\mathbf{g}_i$, we obtain the expression in Theorem 2. $\qquad\square$

#### A.4.2 Extreme Graph SNRs

Let us now turn to the next result, where we analyze the generalization error in the cases where the graph SNR $\Gamma$ takes extreme values.

**Theorem** (Restatement of Theorem 3). *Let $h_\ell^*$ be the classifier from Corollary 1.1, $h_0^*(u) = \mathrm{sgn}(\langle \mathbf{X}_u, \boldsymbol{\mu} \rangle)$ be the Bayes optimal classifier given only the feature information of the root node $u$, and $h_{\mathrm{gcn}}$ be the one-layer vanilla GCN classifier. Then we have that for any fixed $\ell$:*

1. *If $\Gamma = 0$ then $\mathcal{E}(h_\ell^*) = \mathcal{E}(h_0^*) = \Phi(-\gamma)$, where $\Phi$ is the standard Gaussian CDF.*

2. *If $\Gamma \to 1$ then $\xi_\ell \geq 1$ a.s. and $\mathcal{E}(h_\ell^*) \to \mathbf{Pr}(g > \gamma\xi_\ell)$, where $g \sim \mathcal{N}(0,1)$.*

3. *$\mathcal{E}(h_{\mathrm{gcn}}) = \mathbf{Pr}(g > \gamma\xi_1)$.*

*Proof.* Note that when $\Gamma = 0$, i.e., when $a = b$, then $a_k = b_k$ for all $k \in [\ell]$, hence, $c_k = \log(a_k/b_k) = 0$. This implies that all information from the $k$-hop neighbours is truncated to 0 for all $k \in [\ell]$. Thus, the classifier reduces to $h_u^* = h_\ell^*(u, \{\mathbf{X}_v\}_{v \in \eta_\ell(u)}) = \mathrm{sgn}(\psi(\mathbf{X}_u)) = \mathrm{sgn}(g + y_u\gamma) = h_0^*(u, \{\mathbf{X}_u\})$. Thus, the probability that $h_u^* y_u < 0$ is

$$\mathbf{Pr}(h_u^* y_u < 0) = \mathbf{Pr}(y_u g + \gamma < 0) = \Phi(-\gamma),$$

where $\Phi(\cdot)$ is the standard Gaussian CDF.

For the other case where $\Gamma \to 1$, we have two sub-cases: Either $a \to 0$ with $b \neq 0$, or $a \neq 0$ with $b \to 0$. In this case, $c_k \to \infty$ for all $k$, so the classifier takes the form

$$h_\ell^*(u, \{\mathbf{X}_v\}_{v \in \eta_\ell(u)}) = \mathrm{sgn}\left( g + y_u\gamma + \sum_{k \in [\ell]} \sum_{v \in N_k(u)} (g_{k,v} + y_v\gamma) \right).$$

Hence, the probability of making a mistake is

$$\mathbf{Pr}(h_u^* y_u < 0) = \mathbf{Pr}\left( y_u g + \gamma + \sum_{k \in [\ell]} \sum_{v \in N_k(u)} (y_u g_{k,v} + y_u y_v \gamma) < 0 \right)$$

$$= \mathbf{Pr}\left( g > \gamma \frac{|\eta_\ell^{(a)} - \eta_\ell^{(b)}|}{\sqrt{\eta_\ell^{(a)} + \eta_\ell^{(b)}}} \right),$$

where $\eta_\ell^{(a)}, \eta_\ell^{(b)}$ denote the total number of nodes in the $\ell$-neighbourhood $\eta_\ell(u)$ that are in the same class as $u$ and different class as $u$, respectively. The last equation is obtained by using the fact that $(g, \{g_{k,v}\})$ are i.i.d. standard Gaussians. Note that in this case since either $b \to 0$ or $a \to 0$ (but not both), we have $\eta_\ell^{(b)} \to 0$ or $\eta_\ell^{(a)} \to 0$ using (2) for any fixed $\ell$. Thus, $\xi_\ell(a,b) > 1$ a.s. Following a similar analysis, one can find that $\mathcal{E}(h_{\mathrm{gcn}}) = \mathbf{Pr}(g > \gamma \cdot \xi_1(a,b))$. $\qquad\square$

It is interesting to note that we may not have $\xi_1(a,b) > 1$ in general, meaning that a GCN is better than methods that do not use a graph only in the case where $\xi_1(a,b) > 1$.

### A.5   Non-asymptotic Analysis

First, consider the case where $\ell$, the total depth of the neighbourhood is a constant independent of $n$, the number of nodes.

Putting $m = 1$ in Lemma A.1, we see that the probability is bounded by $(8\ell^2(a+b))^{4\ell+1}/n = O(1/n)$. Hence, we conclude that in the limit $n \to \infty$, there are no cycles in any constant-depth neighbourhoods in the graph. In particular, we obtain that the local weak limit $(G, u)$ is a tree.

We now turn to the case where the depth of the neighbourhood is logarithmic in $n$.

**Proposition** (Restatement of Proposition 3.1). *Let $G = (V, E) \sim \mathrm{CSBM}(n, d, \mathbb{P}, \frac{a}{n}, \frac{b}{n})$. Then for any $\ell = c\log n$ such that $c\log(\frac{a+b}{2}) < 1/4$, with probability $1 - O(1/\log^2 n)$, the number of nodes $u \in V$ whose $\ell$-neighbourhood is cycle-free is $n\left(1 - o(\frac{\log^4(n)}{\sqrt{n}})\right)$.*

*Proof.* In Lemma A.2, observe that since $c\log\Delta < 1/4$, we have $\ell < \frac{\log n}{4\log\Delta} = \frac{1}{4}\log_\Delta n$. Thus, putting $m = \log n$, we find that with probability at least $1 - O(1/\log^2 n)$, the number of nodes whose $\ell$-neighbourhood contains at least one cycle is bounded by $O(\log^4(n)\Delta^{2\ell}) = o(\log^4(n)\sqrt{n})$. Hence, the fraction of nodes whose $\ell$-neighbourhood is cycle-free is $1 - o(\frac{\log^4 n}{\sqrt{n}})$. $\qquad\square$

For a fixed node $u \in [n]$, let us denote the number of nodes at distance $k$ (respectively $\leq k$) from $u$ with class label $\pm y_u$ by $U_k^\pm(u)$ (respectively, $U_{\leq k}^\pm(u)$). Also let $n_\pm$ denote the number of nodes with class label $\pm y_u$, so that $n = n_+ + n_-$. Note that $U_0^+(u) = 1, U_0^-(u) = 0$, and conditionally on the sigma-field $\mathcal{F}_{k-1} = \sigma(U_t^\pm(u), t \leq k-1)$, we have

$$U_k^+(u) \sim \mathrm{Bin}\left(n_+ - U_{\leq k-1}^+, 1 - (1-a/n)^{U_{k-1}^+}(1-b/n)^{U_{k-1}^-}\right), \tag{6}$$

$$U_k^-(u) \sim \mathrm{Bin}\left(n_- - U_{\leq k-1}^-, 1 - (1-a/n)^{U_{k-1}^-}(1-b/n)^{U_{k-1}^+}\right). \tag{7}$$

Define $S_k(u) = U_k^+(u) + U_k^-(u)$ to be the number of nodes at distance exactly $k$ from $u$, and denote $\Delta = \frac{a+b}{2}$ to be the expected degree of a node. Correspondingly, recall from (2) that we have

$$\alpha_0 = 1, \beta_0 = 0,$$
$$\alpha_k \sim \mathrm{Poi}\left(\frac{a\alpha_{k-1} + b\beta_{k-1}}{2}\right), \beta_k \sim \mathrm{Poi}\left(\frac{a\beta_{k-1} + b\alpha_{k-1}}{2}\right) \text{ for } k \in [\ell]. \tag{8}$$

Let us now state a useful high-probability bound on $S_k(u) = U_k^+(u) + U_k^-(u)$.

**Lemma A.4.** *[Massoulié, 2014, Theorem 2.3]. For any $\ell = c\log n$ with $c\log\Delta < 1/4$, there exist constants $C, \epsilon > 0$ such that with probability at least $1 - O(n^{-\epsilon})$, $S_k(u) \leq C\Delta^k \log(n)$ for all $u \in [n]$ and all $k \in [\ell]$.*

We now obtain a total variation bound between the sequences $\{U_k^\pm\}_{k\geq 0}$ and $\{\alpha_k, \beta_k\}_{k\geq 0}$.

**Lemma A.5.** *Let $u \in [n]$ be fixed with label $y_u \in \{\pm 1\}$. Let $\ell = c\log n$ with $c\log\Delta < 1/4$. Then the total variation distance between the collections of variables $\{U_k^+(u), U_k^-(u)\}_{k\leq \ell}$ and $\{\alpha_k(u), \beta_k(u)\}_{k\leq \ell}$ is bounded by $O(\log^3 n/n^{1/4})$.*

*Proof.* Define the following events for $C$ as in Lemma A.4:

$$\Omega_k = \{S_k \le C\Delta^k \log n\}, 1 \le k \le \ell. \tag{9}$$

Conditionally on the sigma-field $\mathcal{F}_{k-1} = \sigma(U_t^\pm(u), t \le k-1)$ and the event $\Omega_{k-1}$, we compute the total variation distance between the variables $(U_k^+(u), U_k^-(u))$ and $(\alpha_k(u), \beta_k(u))$. Since $u$ is fixed, we omit it from the notation for brevity. Define the following random variables:

$$W_k^+ \sim \text{Poi}\left(\frac{aU_{k-1}^+ + bU_{k-1}^-}{2}\right), \qquad W_k^- \sim \text{Poi}\left(\frac{aU_{k-1}^- + bU_{k-1}^+}{2}\right).$$

We now apply the Stein-Chen method to bound $d_{\text{TV}}(U_k^\pm, W_k^\pm)$. For more details on this technique, we refer to Stein [1972], Chen [1975], Barbour and Chen [2005]. In particular, we use the fact that for $X_1 \sim \text{Bin}(n, \lambda/n)$, $X_2 \sim \text{Poi}(\lambda)$ and $X_3 \sim \text{Poi}(\lambda')$, $d_{\text{TV}}(X_1, X_2) \le \lambda/n$ and $d_{\text{TV}}(X_2, X_3) \le |\lambda - \lambda'|$. Let us focus on $d_{\text{TV}}(U_k^+, W_k^+)$ as the other case for $d_{\text{TV}}(U_k^-, W_k^-)$ is similar. Construct an intermediate random variable based on the distributions of $U_k^\pm$ as in Eqs. (6) and (7),

$$V_k \sim \text{Poi}\left((n_+ - U_{\le k-1}^+)\left(1 - (1 - a/n)^{U_{k-1}^+}(1 - b/n)^{U_{k-1}^-}\right)\right).$$

Denote $T_t = 1 - \left(1 - \frac{a}{n}\right)^{U_t^+}\left(1 - \frac{b}{n}\right)^{U_t^-}$ for brevity. Note that using triangle inequality,

$$d_{\text{TV}}(V_k, W_k^+) \le \left|(n_+ - U_{\le k-1}^+)T_{k-1} - \frac{aU_{k-1}^+ + bU_{k-1}^-}{2}\right|$$

$$\le \left|\left(n_+ - U_{\le k-1}^+ - \frac{n}{2}\right)T_{k-1}\right| + \left|\frac{aU_{k-1}^+ + bU_{k-1}^- - nT_{k-1}}{2}\right|$$

$$= \left|\left(n_+ - U_{\le k-1}^+ - \frac{n}{2}\right)T_{k-1}\right| + \frac{n}{2}\left|\frac{aU_{k-1}^+ + bU_{k-1}^-}{n} - T_{k-1}\right|$$

$$\le \left|\left(n_+ - U_{\le k-1}^+ - \frac{n}{2}\right)T_{k-1}\right| + \frac{1}{4n}\left(aU_{k-1}^+ + bU_{k-1}^-\right)^2,$$

where in the last inequality we used Fact A.3. Then we obtain the variation distance:

$$d_{\text{TV}}(U_k^+, W_k^+) \le d_{\text{TV}}(U_k^+, V_k) + d_{\text{TV}}(V_k, W_k^+)$$

$$\le T_{k-1} + \left|\left(n_+ - U_{\le k-1}^+ - \frac{n}{2}\right)T_{k-1}\right| + \frac{1}{4n}\left(aU_{k-1}^+ + bU_{k-1}^-\right)^2$$

$$= T_{k-1}\left(1 + \left|n_+ - U_{\le k-1}^+ - \frac{n}{2}\right|\right) + \frac{1}{4n}\left(aU_{k-1}^+ + bU_{k-1}^-\right)^2.$$

Consider now a choice of $c$ such that $c\log\Delta < 1/4$. We have $\ell = c\log n < \frac{1}{4}\log_\Delta n$, implying that $\Delta^\ell \le n^{1/4}$. Recalling (9) corresponding to Lemma A.4, we have that under the event $\Omega_{k-1}$ for $k \le \ell$, the number of nodes at distance $k-1$ is

$$S_{k-1} = U_{k-1}^+ + U_{k-1}^- \le C\Delta^{k-1}\log n \le C\Delta^\ell \log n \le Cn^{1/4}\log n. \tag{10}$$

Observe now that from Fact A.3, $T_{k-1} \le \frac{aU_{k-1}^+ + bU_{k-1}^-}{n}$. Recalling that $y_u$ have a uniform prior, by the Chernoff bound [Vershynin, 2018, Theorem 2.3.1] on $n_+$, we have $|n_+ - \frac{n}{2}| = O(\sqrt{n}\log n)$ with probability at least $1 - 1/\text{poly}(n)$. Thus, we obtain that under this event,

$$d_{\text{TV}}(U_k^+, W_k^+) \le O\left(\frac{|U_{\le k-1}^+|}{n} + \frac{\log n}{\sqrt{n}}\right) \cdot \left(aU_{k-1}^+ + bU_{k-1}^-\right) + \frac{1}{4n}\left(aU_{k-1}^+ + bU_{k-1}^-\right)^2$$

$$\le O\left(\frac{\log n}{\sqrt{n}}\right) \cdot \max(a,b)S_{k-1} + \frac{\max(a,b)^2}{4n}S_{k-1}^2 = O\left(\frac{\log^2 n}{n^{1/4}}\right),$$

where in the last step we used the bound from (10). Now recall that the variables $\{U_k^\pm, \alpha_k, \beta_k\}_{k \in [\ell]}$ are defined as in Eqs. (6) to (8) for all $k \le \ell$. For a fixed $u \in [n]$, we have the base cases $U_0^+ = \alpha_0 = 1$

and $U_0^- = \beta_0 = 0$. Then following an induction argument with a union bound over all $k \in \{1, \ldots, \ell\}$, we have that the variation distance between the sequences $\{U_k^+, U_k^-\}_{k \leq \ell}$ and $\{\alpha_k, \beta_k\}_{k \leq \ell}$ is upper bounded by $O\left(\frac{\log^3 n}{n^{1/4}}\right)$. $\qquad\square$

We now obtain a relationship between the misclassification error on the data model with finite $n$, i.e., $\mathcal{E}_n$ and the error on the limit of the model with $n \to \infty$, i.e., $\mathcal{E}$.

**Theorem** (Restatement of Theorem 4). *For any $1 \leq \ell \leq c \log n$ such that the positive constant $c$ satisfies $c \log(\frac{a+b}{2}) < 1/4$, we have that*

$$\left| \min_{h \in \mathcal{C}_\ell} \mathcal{E}_n(h) - \mathcal{E}_n(h_\ell^*) \right| = O\left(\frac{1}{\log^2 n}\right), \quad \left| \min_{h \in \mathcal{C}_\ell} \mathcal{E}_n(h) - \mathcal{E}(h_\ell^*) \right| = O\left(\frac{1}{\log^2 n}\right).$$

*Proof.* Consider a random feature-decorated graph $G_n \sim \text{CSBM}(n, d, \{\mathbb{P}_\pm\}, a/n, b/n)$, where $\mathbb{P}_\pm$ correspond to the distributions $\mathcal{N}(\pm\boldsymbol{\mu}, \sigma^2 \mathbf{I})$ for the node features given by $\{\mathbf{X}_u\}_{u \in [n]}$. For a classifier $h \in \mathcal{C}_\ell$, the class of all $\ell$-local classifiers, define $\mathcal{E}_n(h)$ to be the probability of misclassification for a uniform at random node $u \in [n]$, i.e., $\mathcal{E}_n(h) = \mathbf{Pr}(y_u \cdot h(u, \{\mathbf{X}_v\}_{v \in \eta_\ell(u)}, \eta_\ell(u)) < 0)$. Since it is known that all classifiers in $\mathcal{C}_\ell$ operate on $u$ given the information in its $\ell$-neighbourhood $\eta_\ell(u)$, we will omit $\eta_\ell(u)$ from the notation and say $h(u)$ instead of $h(u, \{\mathbf{X}_v\}_{v \in \eta_\ell(u)}, \eta_\ell(u))$ when it is understood. Let $\mathbf{P}$ be the joint measure of the variables $\{U_k^\pm\}_{k \leq \ell}$ from Eqs. (6) and (7), and $\mathbf{P}'$ be the joint measure of the variables $\{\alpha_k, \beta_k\}_{k \leq \ell}$ from Eq. (8). Then Lemma A.5 gives us that $d_{\text{TV}}(\mathbf{P}, \mathbf{P}') \leq O\left(\frac{\log^3 n}{n^{1/4}}\right) = o_n(1)$.

Recall $\mathcal{E}(h_\ell^*)$ computed in Theorem 2 for the limiting data model $(G, u)$.

$$\mathcal{E}(h_\ell^*) = \mathbf{Pr}\left[ g + \frac{1}{2\gamma} \sum_{k \in [\ell]} \left( \sum_{i=1}^{\alpha_k} Z_{k,i}^{(a)} + \sum_{i=1}^{\beta_k} Z_{k,i}^{(b)} \right) > \gamma \right]$$
$$= \int \mathbf{Pr}\left[ g + \frac{1}{2\gamma} \sum_{k \in [\ell]} \left( \sum_{i=1}^{\alpha_k} Z_{k,i}^{(a)} + \sum_{i=1}^{\beta_k} Z_{k,i}^{(b)} \right) > \gamma \,\middle|\, \{\alpha_k, \beta_k\}_{k \leq \ell} \right] d\mathbf{P}'.$$

Similarly, we have

$$\mathcal{E}_n(h_\ell^*) = \mathbf{Pr}\left[ g + \frac{1}{2\gamma} \sum_{k \in [\ell]} \left( \sum_{i=1}^{U_k^+} Z_{k,i}^{(a)} + \sum_{i=1}^{U_k^-} Z_{k,i}^{(b)} \right) > \gamma \right]$$
$$= \int \mathbf{Pr}\left[ g + \frac{1}{2\gamma} \sum_{k \in [\ell]} \left( \sum_{i=1}^{U_k^+} Z_{k,i}^{(a)} + \sum_{i=1}^{U_k^-} Z_{k,i}^{(b)} \right) > \gamma \,\middle|\, \{U_k^\pm\}_{k \leq \ell} \right] d\mathbf{P}.$$

Thus, we obtain that

$$|\mathcal{E}_n(h_\ell^*) - \mathcal{E}(h_\ell^*)| \leq d_{\text{TV}}(\mathbf{P}, \mathbf{P}') \leq O\left(\frac{\log^3 n}{n^{1/4}}\right) = o_n(1), \tag{11}$$

Let us now focus on the case with finite $n$. Let $A$ denote the event from Proposition 3.1 where the number of nodes with cycle-free $\ell$-neighbourhoods is $1 - o(\frac{\log^4 n}{\sqrt{n}})$. For a node $u \in G_n$, let $E_u$ denote the event that the subgraph induced by the $\ell$-neighbourhood of $u$, $\eta_\ell(u)$ is a tree. Then observe that for a uniform random node $u \in G_n$,

$$\min_{h \in \mathcal{C}_\ell} \mathcal{E}_n(h) = \mathbf{Pr}(y_u h_{\ell,n}^*(u) < 0)$$
$$= \mathbf{Pr}(E_u) \mathbf{Pr}(y_u h_{\ell,n}^*(u) < 0 \mid E_u) + \mathbf{Pr}(E_u^{\mathsf{c}}) \mathbf{Pr}(y_u h_{\ell,n}^*(u) < 0 \mid E_u^{\mathsf{c}})$$
$$= (1 - o_n(1)) \mathbf{Pr}(y_u h_\ell^*(u) < 0) + o_n(1)$$
$$= \mathcal{E}_n(h_\ell^*) \pm o_n(1).$$

In the above, we used from Proposition 3.1 that $\mathbf{Pr}(E_u) = \mathbf{Pr}(E_u \cap A) + \mathbf{Pr}(E_u \cap A^c) = 1 - O(\frac{1}{\log^2 n})$, and that $\mathcal{E}_n(h_\ell^*) = \min_{h \in \mathcal{C}_\ell} \mathbf{Pr}(y_u h(u) < 0 \mid E_u)$. This establishes the first part:

$$|\min_{h \in \mathcal{C}_\ell} \mathcal{E}_n(h) - \mathcal{E}_n(h_\ell^*)| = O\left(\frac{1}{\log^2 n}\right).$$

Combining the above display with (11), we obtain the second part, i.e.,

$$
\begin{aligned}
\left| \min_{h \in \mathcal{C}_\ell} \mathcal{E}_n(h) - \min_{h \in \mathcal{C}_\ell} \mathcal{E}(h) \right| &= \left| \min_{h \in \mathcal{C}_\ell} \mathcal{E}_n(h) - \mathcal{E}(h_\ell^*) \right| \\
&= \left| \min_{h \in \mathcal{C}_\ell} \mathcal{E}_n(h) - \mathcal{E}_n(h_\ell^*) + \mathcal{E}_n(h_\ell^*) - \mathcal{E}(h_\ell^*) \right| \\
&\leq \left| \min_{h \in \mathcal{C}_\ell} \mathcal{E}_n(h) - \mathcal{E}_n(h_\ell^*) \right| + |\mathcal{E}_n(h_\ell^*) - \mathcal{E}(h_\ell^*)| \\
&= O\left(\frac{1}{\log^2 n}\right) + O\left(\frac{\log^3 n}{n^{1/4}}\right) = O\left(\frac{1}{\log^2 n}\right). \qquad \square
\end{aligned}
$$

## B  Additional Empirical Observations

### B.1  Convergence of Parameters

We observe empirically that gradient descent (`SGD` and `Adam` implementations in the `pytorch` library) converges in the binary setting. In this case, the neural network learns the right parameters corresponding to the parameters of the CSBM, i.e., $\rho$ and $\mathbf{Q}$, such that Architecture 1 realizes the optimal classifier in Corollary 1.1. In Fig. 3, the x-axis denotes the number of epochs elapsed since

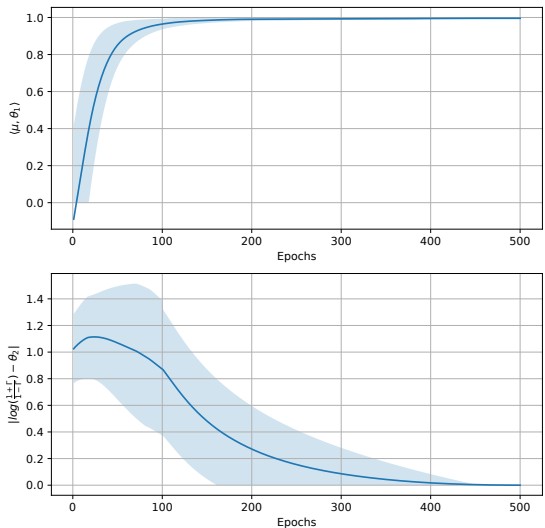

Figure 3: Convergence of parameters of Architecture 1.

the beginning of the training process. For the first plot, the y-axis denotes the cosine similarity between the parameters $\theta_1 \in \mathbb{R}^d$ learned by the MLP $\mathbf{H}^{(L)}$ in Architecture 1 and the ansatz $\boldsymbol{\mu}$ that realizes the optimal classifier; while in the second plot, the y-axis denotes the absolute difference between the clip parameter $\theta_2 \in \mathbb{R}$ and the ansatz value $\log(\frac{1+\Gamma}{1-\Gamma})$. These experiments are performed in the same setting as Fig. 1b with fixed $\Gamma = 0.2$. We see that the parameters converge as the number of training iterations increases. The reported metrics are averaged over 10 trials, and the standard deviation is shown at each iteration using the translucent blue region.

## B.2 Comparison with AMP-BP

We now perform a comparison of our architecture with $\ell = 5$ against the AMP-BP algorithm from [Deshpande et al., 2018] in two different settings. We set $n = 1000$ and work with two values of $d \in \{5, 500\}$. The case $d = 5$ simulates the low-dimensional case where asymptotically $d/n \to 0$, while $d = 500$ represents the high-dimensional case where $d/n \to c$ for a constant $c$. For the AMP-BP algorithm, we choose two values for the number of iterations, $t \in \{5, 20\}$.

Tables 1 and 2 show the results for $\gamma = 1$ with varying values of $\Gamma \in \{0.3, 0.4, 0.5, 0.6\}$. We observe that the classifier obtained after training Architecture 1 almost always outperforms AMP-BP for both low-dimensional and high-dimensional cases.

Table 1: Accuracy metrics for Architecture 1 and AMP-BP for $\gamma = 1$ and $d = 5$.

| Graph signal | 5-local Bayes Optimal | AMP, $t = 5$ | AMP, $t = 20$ |
|---|---|---|---|
| 0.3 | 0.870 | 0.753 | 0.858 |
| 0.4 | 0.954 | 0.816 | 0.890 |
| 0.5 | 0.988 | 0.819 | 0.916 |
| 0.6 | 0.996 | 0.892 | 0.952 |

Table 2: Accuracy metrics for Architecture 1 and AMP-BP for $\gamma = 1$ and $d = 500$.

| Graph signal | 5-local Bayes Optimal | AMP, $t = 5$ | AMP, $t = 20$ |
|---|---|---|---|
| 0.3 | 0.916 | 0.554 | 0.848 |
| 0.4 | 0.995 | 0.558 | 0.877 |
| 0.5 | 0.998 | 0.626 | 0.920 |
| 0.6 | 0.998 | 0.657 | 0.940 |

Tables 3 and 4 show the results for $\gamma = 0.2$, i.e., for a weaker feature signal in the data. Here, we observe that although Architecture 1 outperforms AMP-BP in the low-dimensional regime, it exhibits worse performance than AMP-BP in the high-dimensional regime.

Table 3: Accuracy metrics for Architecture 1 and AMP-BP for $\gamma = 0.2$ and $d = 5$.

| Graph signal | 5-local Bayes Optimal | AMP, $t = 5$ | AMP, $t = 20$ |
|---|---|---|---|
| 0.3 | 0.554 | 0.520 | 0.579 |
| 0.4 | 0.587 | 0.528 | 0.584 |
| 0.5 | 0.823 | 0.604 | 0.756 |
| 0.6 | 0.997 | 0.637 | 0.880 |

Table 4: Accuracy metrics for Architecture 1 and AMP-BP for $\gamma = 0.2$ and $d = 500$.

| Graph signal | 5-local Bayes Optimal | AMP, $t = 5$ | AMP, $t = 20$ |
|---|---|---|---|
| 0.3 | 0.584 | 0.502 | 0.543 |
| 0.4 | 0.706 | 0.525 | 0.694 |
| 0.5 | 0.762 | 0.568 | 0.804 |
| 0.6 | 0.788 | 0.604 | 0.878 |

It is important to note, however, that this is an apples-to-oranges comparison because AMP-BP is not a local algorithm, i.e., the whole graph is visible to the algorithm and all nodes contribute to the classification of every other node. This is not true for Architecture 1, where only the nodes in the $\ell$-hop neighbourhood contribute to this decision. Our notion of optimality is among the class of local algorithms. Furthermore, we observe that AMP-BP with 5 iterations does not converge and obtains a much lower accuracy compared to 20 iterations.

