# OpenReview forum: "Optimality of Message-Passing Architectures for Sparse Graphs"
_NeurIPS.cc/2023/Conference — NeurIPS 2023 poster_

### Official Review · Reviewer_S322 · 2023-07-04

**Soundness:** 3 good
**Presentation:** 3 good
**Contribution:** 3 good
**Rating:** 6
**Confidence:** 4

**Summary:**

This paper studies a specific graph convolutional architecture (and the resultant estimators from an architecture with appropriate learned values)
based in the setting of a contextual stochastic block model: one where a graph with latent clustering also has node features available that
are informative for discovering the clustering. The paper studies this in the "constant average degree and constant feature dimension" regime
and makes the following contributions:

1. Identifies the Bayes-optimal classifier for node clustering restricted to a constant size neighborhood in the graph (using local
weak convergence theory) and shows this matches the GCN architecture proposed.
2. Specializes the classifier to Gaussian features and 2 latent clusters, proving a formula for the bayes optimal classifier
3. Shows that the local classifier is nearly optimal even for GCN architectures of $c \log n$ (for a small enough constant c)
using the locally-tree-like nature of the graph.

**Strengths:**

The main strengths are:

1. Identifies a setting in which specific GCN architectures are provably optimal
2. Uses local weak convergence theory to show some simple but illustrative examples of how (local) Bayes-optimal classifiers
aggregate information across features/graph structure.

**Weaknesses:**

It is hard to get excited about this paper. The main weakness to me is that the Bayes optimal classifier for the tree model is essentially obvious, given Bayes rule. It is far from clear that GCN architectures uninformatively initialized would converge to this, and the paper's experiments do not show this (or in fact consider it from what I could tell). They instead rely on the fact that the architecture they propose can approximate (with some setting of weights) the Bayes optimal classifier. At this point, it is well-known that the success of NN architectures cannot be explained by such existence/approximation theorems.

If the experiments carried out show that naive initializations of the architectures converge to the optimal ones, then that is a priori surprising, and should be discussed more. I might even pivot the paper to start with that as an experimental finding, since the theoretical results are relatively unsurprising.

**Questions:**

Some questions, roughly in order of the paper reading:
1. Why is the notion of SNR $\frac{(a-b)}{a+b}$ when in prior notions of SBM in the constant degree regime it is usually $\frac{(a-b)^2}{2(a+b)}$ which makes sense in terms of the difference of mean / sd of poisson degrees (which is correct in the limit)? If not, how does this reconcile with prior work when features are not meaningful?
2. What happens to Corollary 1.1. when $\Gamma > 1$ (particularly the formula for c_k should be modified somewhat). Naively (using the same formula but with $\Gamma^k -1$), this means that c_k converges to a constant as $k$ diverges. Intuitively, it should be the case that the features should matter less and less as the neighborhood diverges in this regime, irrespective of the snr, which is the behavior for $\Gamma <1$. This suggests that the formula needs more than a naive change for $\Gamma >1$.
3. Same question as 1 on formula (3) and Theorem 3 (the specialization to the Gaussian case)
4. Theorem 3: the statement $\xi_\ell \ge 1$ looks like it needs to be probabilistic statement (almost surely?, with high probability?) or perhaps needs vanishing slack? E.g. take $a$ large, $\ell = 1$ and $b=1$, then there's a constant (but small as a diverges) probability that $\alpha_1$ is 0 and $\beta_1 \leq 2$ is not large so that $\xi_1 <1$.
5. Does the Bayes optimal classifier know the probabilities $Q$ and the densiites? Presumably this is necessary.
6. What about the GCN in Fig 1? Does it learn it from scratch, or initialized at true values, etc?
7. In figure 1, presumably the (roughly) 0.75 accuracy is basically $\Phi(-1)$ at $\Gamma = 0$?

**Limitations:**

The most important weakness is highlighted in the weaknesses.

Second, it is also not obvious how one would change this for dense graphs. Analogous results (though with *very* different proof techniques) might be expected
to hold when graphs have degrees of order $n$ but with within/out cluster probabilities differing by order $1/\sqrt{n}$. How would one identify GCN architectures that might work there?

---

> ### Author Rebuttal · Authors · 2023-08-10
>
> We thank the reviewer for their concise and thought-provoking questions! They really help us develop a better perspective of our contributions and improve our presentation. We address the comments below.
>
> > It is far from clear that GCN architectures uninformatively initialized would converge to this, and the paper's experiments do not show this. They instead rely on the fact that the architecture they propose can approximate (with some setting of weights) the Bayes optimal classifier.
>
> We would like to clarify that our experiments indeed show that in the binary case, Architecture 1 uninformatively initialized converges to the ansatz, i.e., it is able to learn the right values of $Q$ and $\rho$. Both Figures 1 and 2 depict the test accuracy of trained models that were initialized uninformatively (uniformly at random). We have added this information in the supplementary material, and also **attached a pdf of the plots showing the convergence in the general rebuttal response** above.
>
> > If the experiments carried out show that naive initializations of the architectures converge to the optimal ones, then that is a priori surprising. I might even pivot the paper to start with that as an experimental finding, since the theoretical results are relatively unsurprising.
>
> We do indeed observe that naive initializations of the architecture converge to the optimal one in the binary Gaussian case (see the attached figures). We agree that this is a priori surprising, and we will add a discussion about this in the revision. However, we humbly request the reviewer to consider that pivoting the paper to start with this experimental finding will change the primary focus of our work, which is to show that the optimal estimator in the regime we study is realizable by a message-passing GNN architecture.
>
> > Why is the notion of SNR $\frac{|a-b|}{a+b}$ when in prior notions of SBM in the constant degree regime it is usually $\frac{(a-b)^2}{2(a+b)}$ which makes sense in terms of the difference of mean / sd of poisson degrees? How does this reconcile with prior work when features are not meaningful?
>
> When computing the optimal classifier, $\Gamma=\frac{|a-b|}{a+b}$ is what naturally shows up in the estimator's expression. We agree that in previous works where features are not present, the SNR is defined as $\frac{(a-b)^2}{2(a+b)}$. We believe that this is because of the nature of the previous work on detectability thresholds for SBMs. It is known (Mossel et al, Massoulie et al) that weak recovery is possible if and only if this SNR > 1, however, we do not study detectability thresholds. Instead, we compute the optimal architecture in terms of the parameters of the data model, and the quantities that naturally pop up in the expression of the estimator are what we interpret as signals in the data.
>
> > What happens to Corollary 1.1 when $\Gamma>1$. Naively (using the same formula but with $\Gamma^k - 1$), this means that c_k converges to a constant as $k$ diverges. Intuitively, it should be the case that the features should matter less and less as the neighborhood diverges in this regime, irrespective of the snr, which is the behavior for
> $\Gamma < 1$. This suggests that the formula needs more than a naive change for $\Gamma>1$.
>
> We would like to note that $\Gamma=\frac{|a-b|}{a+b}$ is always $\le1$ for $a,b\ge 0$. Hence, we are not concerned with the case $\Gamma>1$. However, the reviewer's intuition overall is exactly what we intended to communicate.
>
> > Theorem 3: the statement $\xi_\ell\ge 1$ looks like it needs to be probabilistic statement (almost surely?, with high probability?) or perhaps needs vanishing slack?
>
> Thank your for pointing this out. The statement should say $\xi_\ell>1$ a.s. We will fix this in the revision.
>
> > Does the Bayes optimal classifier know the probabilities $Q$ and the densiites? Presumably this is necessary.
>
> We believe that this relates to the first comment about the architecture's parameters converging to the optimal parameters with gradient descent. As we mentioned above, the architecture when trained on CSBM data indeed learns the right values of $Q$ and $\rho$. It does not know the right values in advance for the experiments we show in the paper.
>
> > What about the GCN in Fig 1? Does it learn it from scratch, or initialized at true values, etc?
>
> All the architectures in fig 1 (MLP, GCN, Architecture 1) are trained from scratch with uniform random initializations. We have added this information in the revision. Thank you for pointing this out, it helps us clarify that our results are indeed as strong as one would ideally expect.
>
> > In figure 1, presumably the (roughly) 0.75 accuracy is basically $\Phi(-1)$ at $\Gamma=0$?
>
> We believe the reviewer wanted to say that the accuracy is roughly $0.85$, which is basically $\1 - \Phi(-1) = \Phi(1)$ at $\Gamma=0$. Yes, this is the correct interpretation.
>
> > it is also not obvious how one would change this for dense graphs. Analogous results (though with very different proof techniques) might be expected to hold when graphs have degrees of order $n$, but with within/out cluster probabilities differing by order $1/\sqrt{n}$. How would one identify GCN architectures that might work there?
>
> This is a very interesting question, and it is true that our current technique is limited to sparse graphs. We rely on the fact that for the sparse regime that we study, a substantial fraction of nodes in the graph have locally tree-like neighbourhoods up to depth $c\log n$ for a suitably bounded $c$. This fails for dense graphs and make it difficult to compute an informative expression for the optimal architecture. By 'informative', we mean an expression that is well-defined in terms of recognizable signals in the data. We leave the question of extending our results to dense graphs as an interesting open problem and a challenging direction for future work.

---

> > ### Comment · Reviewer_S322 · 2023-08-21
> > **Thanks for the detailed response**
> >
> > First, thank you to the authors for the detailed response.
> > ```
> > We would like to clarify that our experiments indeed show that in the binary case ... i.e., it is able to learn the right . Both Figures 1 and 2 depict the test accuracy of trained models that were initialized uninformatively (uniformly at random). We have added this information in the supplementary material, and also attached a pdf of the plots showing the convergence in the general rebuttal response
> > ```
> > Thanks, this is quite interesting behavior and non-obvious. Please make sure to mention this in the final version!
> >
> > ```
> > We would like to note that Gamma is always < 1 ...
> > ```
> > Whoops, my mistake. thanks for correcting.
> >
> > ```
> > When computing the optimal classifier...
> > ```
> > This is interesting, because one could always reduce your model to the standard SBM by taking uninformative features (i.e. $\rho_+ = \rho_-$ in the example). This yields an uninformative guess from the max-belief propagation, which is expected because there is nothing to break the symmetry. This implies that you need $\rho_+ \neq \rho_-$ for a non-trivial result. It might suffice even that they be $\varepsilon_n$ different in some appropriate metric provided $\varepsilon_n \to 0$ slowly enough . This is standard weakness of belief propagation (also is present in ref 1, though spectral methods get rid of this, and one can imagine a non-asymptotic analysis to be possible).
> >
> > Raising my score to 6.

---

### Official Review · Reviewer_K3Xe · 2023-07-05

**Soundness:** 4 excellent
**Presentation:** 4 excellent
**Contribution:** 3 good
**Rating:** 7
**Confidence:** 4

**Summary:**

The paper considers the inference problem of node classification in feature-decorated locally tree-like sparse graphs. The authors introduce and motivate a notion of asymptotic local Bayes optimality for local estimators. The first main result, Theorem 1, concerns the characterisation of the Bayes optimal (in the former sense) classifier in the setting considered. This theorem implies that this estimator is implementable in a message passing graph neural network architecture. The authors then consider the case in which the features are Gaussian. In this context, Theorem 2 establishes the asymptotic local Bayes optimal error while Theorem 3 compares this optimal classifier with other standard classifiers in some extreme signal-to-noise settings. These results are contrasted with numerical experiments. Finally, Theorem 4 gives a finite size bound for the difference between the misclassification error of the estimator proposed and the minimum misclassification error of local classifiers.


**Strengths:**

- The questions addressed in the paper are relevant, interesting, and are part of an active research field.
- The results are interesting, rigorous, and thorough and the proofs are clear.
- The inclusion of finite size bounds for the difference between the performance of the classifier proposed and the optimal performance is of particular interest and renders the analysis presented very complete.

**Weaknesses:**

- Maybe the most relevant weakness of the work is, in my opinion, that it solely focuses in locally tree-like graphs. Although this kind of models are very extended in the literature, the topologies present in many of the applications mentioned are not expected to be of this kind. Indeed, in many social settings, for example, the kind of networks that are observed have diverse clustering coefficient values. Though the paper undoubtedly has theoretical value, this will probably limit the impact it could have in more applied research communities.

**Questions:**

General comments:
- The analytical tools used require the graphs to be locally tree-like and extending the analysis to non locally tree-like graphs is clearly beyond the scope of the work. But it is my opinion that it would make the case for the paper much stronger if the authors could add some numerical simulations comparing the classifier proposed and other estimators in some non locally tree-like settings. This would address to some extent the comment on the Weaknesses' section above.
- The analysis presented focuses mainly in the case of two classes. This is understandable as it renders the presentation more clear. However, it would be good if the authors could add a brief discussion on how the complexity of the estimator depends with respect to c. Is it feasible to compute it for reasonable graph sizes and a moderately large number of classes?

Some particular comments:
- In line 270 after the dot the next word should be capitalised.
- In Figure 1(a), is the value of $\Gamma$ used above the transition? I guess it is but it would be good if this is explicitly stated.
- The convergence rate of Theorem 4 is rather slow. It would thus be specially interesting to have some estimation of the constant for this bound. Do the authors think that this would be possible by some adaptations of the proof?

**Limitations:**

Although the limitations of the work are not explicitly stated, the context and reach of the results are clear in the presentation.

---

> ### Author Rebuttal · Authors · 2023-08-10
>
> We thank the reviewer for their positive review and encouraging comments about our work! We address the questions below.
>
> > Maybe the most relevant weakness of the work is, in my opinion, that it solely focuses in locally tree-like graphs. Although this kind of models are very extended in the literature, the topologies present in many of the applications mentioned are not expected to be of this kind. Indeed, in many social settings, for example, the kind of networks that are observed have diverse clustering coefficient values. Though the paper undoubtedly has theoretical value, this will probably limit the impact it could have in more applied research communities.
>
> Thank you for raising this important concern. We agree that the analysis in our paper is limited to locally tree-like graphs, since we look at the regime of constant degree. Although the architecture we obtained can be evaluated on denser graphs as well, it may not necessarily be optimal in that regime. We believe that our work will inspire further research in this direction, where the quest for optimal architectures is pursued for other topologies suited to many other applications. We will add a discussion about this in the conclusion section of the paper.
>
> > The analytical tools used require the graphs to be locally tree-like and extending the analysis to non locally tree-like graphs is clearly beyond the scope of the work. But it is my opinion that it would make the case for the paper much stronger if the authors could add some numerical simulations comparing the classifier proposed and other estimators in some non locally tree-like settings. This would address to some extent the comment on the Weaknesses' section above.
>
> We agree with the reviewer that extending the analysis to locally tree-like graphs is beyond the scope of the current work, and is a very interesting potential future direction of research. We believe that a comparative study of this architecture against other baselines on graphs that are not locally tree-like would be very interesting and would make the architecture more appealing, and we thank the reviewer for this suggestion. We omitted this in the current work as our primary focus (and the main result, Theorem 1) is for the regime where the average degree is constant and $\ell<c\log n$ for a suitably bounded constant $c$.
>
> > The analysis presented focuses mainly in the case of two classes. This is understandable as it renders the presentation more clear. However, it would be good if the authors could add a brief discussion on how the complexity of the estimator depends with respect to c. Is it feasible to compute it for reasonable graph sizes and a moderately large number of classes?
>
> Our implementation and comparison with MLP and GCN is done for the binary case, however, our main result (Theorem 1) and the architecture we describe (Architecture 1) are for the general case of multiple classes. The estimator has a nice closed form and the architecture makes the estimator realizable. The preprocessing step is a bit computationally expensive for very large graphs for the case of multiple classes due to the calculation of non-backtracking walk matrices. However, we believe that further research in this direction will improve on our preprocessing and help us realize this estimator more efficiently for large graphs.
>
> > In Figure 1(a), is the value of $\Gamma$ used above the transition? I guess it is but it would be good if this is explicitly stated.
>
> The plots in Figure 1 show the accuracy of three different neural networks (MLP, GCN, and Architecture 1) trained on the CSBM. The value of $\Gamma$ is fixed for fig 1 (a), and we plot the accuracy is plotted for a test set on the same distribution but previously unseen features and graph.
>
> > The convergence rate of Theorem 4 is rather slow. It would thus be specially interesting to have some estimation of the constant for this bound. Do the authors think that this would be possible by some adaptations of the proof?
>
> We agree that the convergence rate $O(1/\log^2 n)$ is very slow. For an estimation of the constant, we refer to L. Massoulié. Community Detection Thresholds and the Weak Ramanujan Property. In Proceedings of the Forty-Sixth Annual ACM Symposium on Theory of Computing, page 694–703, 2014. Unpacking Lemma 4.2 in the reference, we see that the constant depends on many other constants that are used throughout its proof, and one can choose suitable values for these constants to arrive at a constant $<2$.

---

> > ### Comment · Reviewer_K3Xe · 2023-08-14
> >
> > I would first like to thank the authors for their detailed response.
> >
> > > Thank you for raising this important concern. We agree that the analysis in our paper is limited to locally tree-like graphs, since we look at the regime of constant degree. Although the architecture we obtained can be evaluated on denser graphs as well, it may not necessarily be optimal in that regime. We believe that our work will inspire further research in this direction, where the quest for optimal architectures is pursued for other topologies suited to many other applications. We will add a discussion about this in the conclusion section of the paper.
> >
> > Okay. So your point is that the motivation for the setting considered is not that it is interesting for applications per se. But rather that the ideas contained could motivate further research in settings that are interesting for applications. Is this right? I would appreciate if this is discussed in the revised version.
> >
> > > We agree that the convergence rate $\mathcal{O}(1/\log^2(n))$ is very slow. For an estimation of the constant, we refer to L. Massoulié. Community Detection Thresholds and the Weak Ramanujan Property. In Proceedings of the Forty-Sixth Annual ACM Symposium on Theory of Computing, page 694–703, 2014. Unpacking Lemma 4.2 in the reference, we see that the constant depends on many other constants that are used throughout its proof, and one can choose suitable values for these constants to arrive at a constant $<2$.
> >
> > Thanks for the clarification. This answers my concern. It would be interesting if this were to be mentioned in the revised version.
> >
> > I would also add that I share the opinion of the area chair that comparison with previous message-passing algorithms for large dimensional features would enrich the manuscript. I would be interested to see how this comparison pays out.

---

> > > ### Author Response · Authors · 2023-08-19
> > > **Further response and thanks**
> > >
> > > > Okay. So your point is that the motivation for the setting considered is not that it is interesting for applications per se. But rather that the ideas contained could motivate further research in settings that are interesting for applications. Is this right? I would appreciate if this is discussed in the revised version.
> > >
> > > Thank you for the suggestion. This will help us clarify the objective and motivate further research! We agree and will include a discussion on this aspect in the revision. Our work is also partly in response to a line of work attempting to design and study GNN architectures that go beyond message-passing, which we mention in the introduction and related works sections.
> > >
> > > > I would also add that I share the opinion of the area chair that comparison with previous message-passing algorithms for large dimensional features would enrich the manuscript. I would be interested to see how this comparison pays out.
> > >
> > > As suggested, we performed these experiments by implementing the AMP algorithm in Deshpande et al. 2018. Please see our general response for details on this.

---

### Official Review · Reviewer_mQCx · 2023-07-06

**Soundness:** 3 good
**Presentation:** 2 fair
**Contribution:** 4 excellent
**Rating:** 7
**Confidence:** 5

**Summary:**

The authors propose a local analysis of the CSBM. They define and derive a local Bayes-optimal classifier and show it can be achieved by a message-passing architecture. They give an interpretation of it in the limiting cases when the graph caries no or all information and support it with numerical experiments.


**Strengths:**

The topic of this article is of great interest. CSBM has been widely used as an artificial dataset and deriving an optimal GNN for it would be good. For this reason I think this article should be published.

**Weaknesses:**

1 In the CSBM usually one considers the high-dimensional limit where the dimension d of the features proportional to n, to model what happens in ML. Here the authors consider d=O(1), which seems limited. I do not see if this is just for convenience or if it is a stronger limitation to this work. For the high-dimensional Gaussian mixture Bayes-optimal classifiers are known and the MLP the authors use works. Could the authors better explain this point?

2 A weakness of this article is a lack of clarity:

– the analysis the authors give is local, the graph is tree-like. For l ~ O(log n) it will not hold. I was a bit confused; maybe the authors should emphasize that l is fixed compared to n, that there are no loops and many things are independent;

– in the definition of architecture 1, the authors should precise that the l and L of the first line (for H^(l)) have nothing to do with the calligraphic l of the second line (the size of the neighborhood). L. 121 "as a simple MLP" –> "as the output of a simple L-layer MLP";

– maybe introducing the model before theorem 1 is counter-intuitive. The authors could explain that the model is a way to realize theorem 1, that the NN just learns rho and Q. This would ease the introduction of tilde A;

– the experiments of fig. 1 and 2 are not clear. What are the training procedure, l, L, training nodes, number of runs, ...

3 It is a pity that the numerical experiments deal with the interpolation and not the major point of the article: the optimality. The authors could compare against other GNNs. Also one has access to the conjecturally-optimal performances on CSBM (in Deshpande '18 for instance); how far is this l-neighborhood model? does the limit large l converge to these or is there a gap (due to the cycles)?

4 How does this model deal with train labels? In architecture 1 the authors assume there is no train node in the l-neighborhood of u, no? otherwise the Bayes-optimal classification would take the labels in account, M_{u,i} = (H_{u,j=label of u} + log Q_{i,j=label of u}). The prior distribution of node labels is not iid uniform in the semi-supervised setting.

I would be pleased to give a higher rating to this article if the authors improve or comment on these points. If the authors develop them in their revised manuscript maybe they could summarize part 3.5.

**Questions:**

Some more general questions:

1 As to the training: can the performances of the trained model be compared to theorem 2? do the learned Q and rho match the ones of the binary CSBM?

2 Does this neural network generalize well to other datasets?

3 Would the authors have an idea how to generalize their results to non-local large l estimators?

Typos :

l. 150 "for a class of estimators this general," ?

ref. l. 409 Andrea Montanari is missing.

**Limitations:**

See above.

---

> ### Author Rebuttal · Authors · 2023-08-09
>
> We thank the reviewer for their encouraging review and insightful comments that helped us improve the clarity and credibility of our work. We address these below.
>
> > In the CSBM usually one considers the high-dimensional limit. the authors consider d=O(1) which seems limited.
>
> We agree that the high-dimensional limit is usually considered where $d$ is proportional to $n$ in the CSBM literature. However, in popular benchmark datasets for node classification, the number of features ($d$) is relatively very small compared to $n$; see for example, OGB datasets: products (n=2.4m, d=100), proteins (n=130k, d=8), papers100m (n=100m, d=128). Therefore, it is not very clear that the high-dimensional scaling is the relevant one. The existing literature on CSBM is focused on $d$ being proportional to $n$, while we consider the setting with fixed $d$. We added a discussion about this in the revision.
>
> > the analysis is for locally tree-like graphs. For $\ell = O(\log n)$ it will not hold...
>
> Although the locally tree-like property will not hold for $\ell= O(\log n)$ in general, in Proposition 3.1 we show that if $\ell<c\log n$ for a bounded $c$, then with high probability the graph is still locally tree-like. We also show in Theorem 4 that for this setting of $\ell$, even in the non-asymptotic case the performance of the classifier $h^{*}$ is close to that of the optimal classifier ${\rm argmin}\_{h\in\mathcal{C_\ell}}\\,\mathcal{E}_n(h)$. Thus, although we do not consider the entire regime where $\ell= O(\log n)$, we do consider the case beyond fixed $\ell$.
>
> > Definition of arch 1: l and L of the first line have nothing to do with the calligraphic l.
>
> Thank you! We incorporated this in the revision.
>
> > The authors could explain that the model is a way to realize theorem 1...
>
> We understand that it may seem reasonable to introduce the model after Theorem 1 from a theoretical perspective, but we also wanted to show the message-passing architecture before a theoretical analysis, and then motivate the architecture using Theorem 1. We discuss right after Theorem 1 about Architecture 1 being a way to realize the classifier in Theorem 1.
>
> > What are the training procedure, l, L, training nodes, number of runs...
>
> The plots are for $\ell=2$ (two-hop neighbourhoods) and $L=1$ (single-layer MLP). Our experiments with $\ell,L\in\{1,2,3\}$ yield the same result in terms of the plots. The networks are trained on a dataset with 10k nodes, and tested on another dataset with 10k nodes, plotting the average test accuracy across 50 trials. We added this in the revision.
>
> > the numerical experiments deal with interpolation and not optimality. The authors could compare against other GNNs and the conjecturally-optimal performances on CSBM
>
> We aimed to show optimality to some extent through the interpolation. We find that with strong graph signal, the optimal classifier mimics a GCN, while with weak graph signal, it mimics an MLP disregarding the noisy graph. This interpolation showcases model optimality across signal strengths, ranging from MLP to GCN extremes and surpassing both throughout. The conjecturally optimal performance on CSBM is for the case where $d$ is proportional to $n$, and our regime of study does not consider this setting. As suggested by the reviewer in an earlier comment, we have added a discussion about this in the revision.
>
> >  how far is the l-neighborhood?...
>
> We take $\ell$ to be as far as $c\log n$ for a suitably bounded constant $c$, as mentioned in Proposition 3.1 and Theorem 4. Beyond this limit, we are not able to guarantee that the $\ell$-neighbourhoods of a substantial fraction of nodes in the graph are cycle-free with high probability, hence we do not consider larger values of $\ell$.
>
> > In architecture 1 the authors assume there is no train node in the l-neighborhood of u, no? otherwise the Bayes-optimal classification would take the labels in account. The prior distribution of node labels is not iid uniform in the semi-supervised setting.
>
> We agree that the architecture doesn't assume labels for the $\ell$-neighbourhood of $u$. The problem's objective is: Given a node's features and its $\ell$-neighbourhood, predict its label. The learning process computes gradients using only node $u$'s training label, despite the output considering features of all $\ell$-neighbourhood nodes. Although the prior label distribution might not be iid uniform in the semi-supervised setup, introducing this complexity alters the Bayes classifier, complicating analysis. Nonetheless, generalizing to this scenario isn't challenging, and the model would adapt to learn the appropriate node label distribution in that case.
>
>
> > Can the performances of the trained model be compared to theorem 2? do the learned Q and rho match the ones of the binary CSBM?
>
> Yes! Our plots in fig 1,2 are where randomly initialized models were trained. The learned $Q$ and $\rho$ match the optimal value for the binary CSBM. We have added this information in the supplementary document now with plots showing convergence of model parameters to the right $Q$ and $\rho$ against the training iterations. **These plots can be found in our rebuttal one-page pdf response**.
>
> > Does it generalize to other datasets?
>
> We evaluate the model on a completely different dataset (but with the same distribution) than the one it is trained on. Figures 1 and 2 are plots of the performance on these unseen datasets.
>
> > Can results generalize to non-local large l estimators?
>
> Thanks for this important question! We are currently limited in this regard because to increase $\ell$ beyond $c\log n$, we need mathematical tools that can deal with the large amount of correlations in the data due to the presence of a large number of cycles in the neighbourhoods of a non-diminishing fraction of nodes. We leave this as a very interesting direction for future work.
>
> Thank you for pointing out the typos. We have fixed them in the revision.

---

> > ### Comment · Reviewer_mQCx · 2023-08-14
> >
> > > The plots are for $\ell=2$ (two-hop neighbourhoods) and $L=1$ (single-layer MLP). Our experiments with $\ell,L\in{1,2,3}$ yield the same result in terms of the plots. The networks are trained on a dataset with 10k nodes, and tested on another dataset with 10k nodes, plotting the average test accuracy across 50 trials. We added this in the revision.
> >
> > Thanks for the details. I did not understand that the test was on a new graph sampled from the CSBM. A more common and realistic setting is the semi-supervised setting, when one has only one graph and a fraction of its labels are revealed (eg PubMed, Cora or OGB arxiv:2005.00687). The authors should precise this when introducing the model.
> >
> > Could the authors precise what a training batch consists in? does the network see all the node labels of the train graph?
> >
> > >>    In architecture 1 the authors assume there is no train node in the l-neighborhood of u, no? otherwise the Bayes-optimal classification would take the labels in account. The prior distribution of node labels is not iid uniform in the semi-supervised setting.
> >
> > > We agree that the architecture doesn't assume labels for the $\ell$-neighbourhood of $u$. The problem's objective is: Given a node's features and its $\ell$-neighbourhood, predict its label. The learning process computes gradients using only node $u$'s training label, despite the output considering features of all $\ell$-neighbourhood nodes. Although the prior label distribution might not be iid uniform in the semi-supervised setup, introducing this complexity alters the Bayes classifier, complicating analysis. Nonetheless, generalizing to this scenario isn't challenging, and the model would adapt to learn the appropriate node label distribution in that case.
> >
> > If the test nodes and the train nodes are not connected then my comment has less importance: the classifier cannot directly use the train labels to predict the test label.
> >
> > However, in the semi-supervised setting, to call this a Bayes-optimal classifier would be a bit misleading: one could do better simply taking in account the given train labels.
> >
> > > Yes! Our plots in fig 1,2 are where randomly initialized models were trained. The learned $Q$ and $\rho$ match the optimal value for the binary CSBM. We have added this information in the supplementary document now with plots showing convergence of model parameters to the right $Q$ and $\rho$ against the training iterations. These plots can be found in our rebuttal one-page pdf response.
> >
> > Ok, thanks.
> >
> > >> Does it generalize to other datasets?
> >
> > > We evaluate the model on a completely different dataset (but with the same distribution) than the one it is trained on. Figures 1 and 2 are plots of the performance on these unseen datasets.
> >
> > When turning the classifier of theorem 2 into a neural network, one expected advantage would be to make it more robust, more able to generalize, eg on CSBM with different parameters.
> >
> > ---
> > Overall, this article is interesting, and worth a publication I think, because it proposes a way to derive an optimal (in a particular sense) GNN. It raises many questions.
> >
> > A limitation, as pointed out by reviewer S322, is that a classifier on a tree has little interest. Also, the properties of the resulting network seem interesting but they are not well studied.

---

> > > ### Author Response · Authors · 2023-08-19
> > > **Further response and thanks**
> > >
> > > > Thanks for the details. I did not understand that the test was on a new graph sampled from the CSBM. A more common and realistic setting is the semi-supervised setting, when one has only one graph and a fraction of its labels are revealed (eg PubMed, Cora or OGB arxiv:2005.00687). The authors should precise this when introducing the model.
> > >
> > > Thank you! We have conducted further experiments for the semi-supervised setting, and as expected from a theoretical standpoint, we obtain precisely the same results. This is expected because given that our dataset is synthetic, there is no difference between the distributions of an unseen node in the same graph as the training set or a new node in a completely unseen graph with the same distribution.
> > >
> > > > Could the authors precise what a training batch consists in? does the network see all the node labels of the train graph?
> > >
> > > In the current set of experiments, this is true. The network sees all node labels in the train graph, but the test graph is completely new. As the reviewer says later, the classifier cannot use the labels of the train graph directly. However, we also did experiments suggested by the reviewer with the semi-supervised setting and obtained similar results. We will include a discussion in the revision for clarity.

---

### Official Review · Reviewer_LT99 · 2023-07-11

**Soundness:** 3 good
**Presentation:** 3 good
**Contribution:** 2 fair
**Rating:** 4
**Confidence:** 3

**Summary:**

The focus of this research paper was the investigation of node classification problem on sparsely populated graphs that exhibit local tree-like structures. Thy introduced asymptotic local Bayes optimality to define the ideal performance standard for node classification tasks. By utilizing this criterion, the paper derived the optimal classifier for a wide range of statistical data models with diverse distributions of node features and edge connectivity. The research paper further assessed the generalization error of this classifier and conducted a theoretical comparison of its performance with existing learning methods. This evaluation was carried out on a model that inherently possesses identifiable signal-to-noise ratios (SNRs) in the data.



**Strengths:**

* Existing works explored conventional message-passing graph neural network architectures. However, their analyses heavily depend on two key assumptions: firstly, the graph is not excessively sparse, and secondly, the node features are represented as a Gaussian mixture. In contrast, this study investigates the realm of highly sparse graphs, where the expected degree of a node is on the order of O(1). Moreover, it considers nodes that extend beyond the immediate neighbors, encompassing nodes at fixed distances.

* Their result holds for a general multi-class statistical model with arbitrary continuous or discrete feature distributions and arbitrary edge-connectivity probabilities between all pairs of classes.

* They showed that in scenarios where the graph signal-to-noise ratio (SNR) is extremely low, the architecture simplifies to a basic MLP that disregards the underlying graph structure. Conversely, when the SNR is significantly high, their architecture transforms into a standard convolutional network that aggregates information from all nodes within the local neighborhood. However, in the intermediate SNR regime, it exhibits interpolation behavior and outperforms both the simple MLP and the typical graph convolutional network (GCN).




**Weaknesses:**

* The analysis in the paper seems interesting but more experimental results are required to support the claims. For example, comaprison with the baselines on existing benchmarks in GCN literature as well as more synthetic types of graphs (e.g scale-free, random etc) with controllable degree distribution to enforce different level of sparsity.


* The proposed method relies on the pre-processing step to calculator A^\~(k) that models a non-backtracking walk of length k that considers new nodes in the distance-k neighborhood that were not discovered. Specifically A\~(k)uv=1 If and only if v is present in the distance k neighborhood of u but not within the distance (k-1) neighborhood. Such a calculation seems to be expensive and might not be scalable. Would you please elaborate on that?


* Q^k models the probability of observing a distance k path between a pair of nodes in two classes. I am wondering why the log (Q^k) is considered in calculation of M. What is the intuition? What would happen if the Q^k has been used? Having abolition study would help in understanding.

* How is the performance of proposed approach in compare to the traditional GCN baselines in real datasets? The papers just considered synthetic graphs with controllable degree but more experimental results are needed on exiting benchmarks. Also for the synthetic graph what type of graph is it and what is the underlying distribution?



**Questions:**

* Analysis of the cost associated with the preprocessing step and discussion on scalability of the proposed approach

* What is impact of log(Q^k) vs Q^k on the performance?

Please look at the weakness for more detailed question.

**Limitations:**

The paper discussed the limitation of the proposed approach.

---

> ### Author Rebuttal · Authors · 2023-08-09
>
> We thank the reviewer for their helpful comments and questions. We address the comments below.
>
> > The analysis in the paper seems interesting but more experimental results are required to support the claims. For example, comaprison with the baselines on existing benchmarks in GCN literature as well as more synthetic types of graphs (e.g scale-free, random etc) with controllable degree distribution to enforce different level of sparsity.
>
> We sincerely request the reviewer to consider that our claims are stated as theorems and are rigorously proved in the paper. We agree that more experimental results will help demonstrate our results more, but in our opinion, they do not provide more insight unless we change the setting of the experiments substantially (one example would be what the reviewer suggested: scale-free or power-law degree-distributed graphs). We note that the primary focus of our work is to show that the optimal GNN architecture for the CSBM with arbitrary feature distributions is a message-passing architecture. We believe that this is a very important result in itself, and decided to leave similar results on many other random graph models (other than the stochastic block model) as future work. Since our theorems specifically rely on the graphs being sparse enough so that $\ell=c\log n$ depth neighbourhoods for a substantially large number of nodes are tree-like, we do not consider dense graphs for experiments.
>
> > The proposed method relies on the pre-processing step to calculator A^(k) that models a non-backtracking walk of length k that considers new nodes in the distance-k neighborhood that were not discovered. Specifically A^(k)_{uv}=1 If and only if v is present in the distance k neighborhood of u but not within the distance (k-1) neighborhood. Such a calculation seems to be expensive and might not be scalable. Would you please elaborate on that?
>
> We completely agree with the reviewer that the pre-processing step is expensive and may not be scalable for extremely large graphs with more than tens of millions of nodes. However, there exist neighbour-sampling techniques that can be explored as a potential future direction of work to make the architecture practically more useful. The primary scope of our paper is to introduce a message-passing architecture that is provably optimal for a very general statistical data model, and thus, we decided to leave the study of computational pre-processing costs as potential future work. We sincerely believe that further research will be able to construct more efficient pre-processing techniques for implementing this architecture for large-scale graphs.
>
> > Q^k models the probability of observing a distance k path between a pair of nodes in two classes. I am wondering why the log (Q^k) is considered in calculation of M. What is the intuition? What would happen if the Q^k has been used? Having abolition study would help in understanding.
>
> The $\log Q^k$ shows up in the architecture because of the maximization of log-likelihood. Our goal was to show that the optimal architecture in the regime we study is a message-passing architecture. Message-passing architectures aggregate messages from all the nodes in the neighbourhood to classify a node of interest. The distribution of the neighbourhood can be expressed as a product of probabilities, and taking a $\log$ helps consider the distribution in terms of a sum of log probabilities. We could have used $Q^k$ instead of $\log Q^k$ in the architecture, but then the message-passing mechanism will need to consider products instead of sums of messages from the nodes.
>
> > How is the performance of proposed approach in compare to the traditional GCN baselines in real datasets? The papers just considered synthetic graphs with controllable degree but more experimental results are needed on exiting benchmarks. Also for the synthetic graph what type of graph is it and what is the underlying distribution?
>
> Thank you for this important question! The main objective of our paper is to establish the theoretical foundation and principles behind the optimal message-passing architecture. We humbly emphasize that our goal is to provide insights about message-passing that could be applied across various domains, and conducting experiments on real datasets is an interesting future endeavour that could follow our theoretical work. We define the synthetic data model that we experiment with, in detail in Sections 3.2 (general case) and 3.4 (binary case and Gaussian case), where we describe the underlying distributions of both the graph and the node features.

---

### Author Rebuttal · Authors · 2023-08-09

In this comment, we respond to the general comments and questions of the reviewers about the convergence of the architecture's parameters, referred to as $(\theta_1, \theta_2)$ in the attached pdf, to the optimal values associated with $\rho$ and $Q$. We emphasize that figures 1 and 2 in the paper are indeed for a network that is initialized uniformly at random, and then trained using a CSBM dataset. In the attached pdf of this rebuttal, we show two plots showing the convergence of the parameters of the architecture to the optimal values associated with $\rho$ and $Q$, i.e., $\mu$ and $\log(\frac{1+\Gamma}{1-\Gamma})$. In the first plot, we show that the normalized inner product between $\theta_1$ and $\mu$ converges to $1$ as the number of iterations increase. Similarly in the second plot we show that $\theta_2$ converges to $\log(\frac{1+\Gamma}{1-\Gamma})$ as the number of iterations increase (here we show that the absolute difference between the values goes to 0). The settings of these plots are the same as that in Fig 1b with $\Gamma=0.2$.

We address all other comments and questions individually for each reviewer.

---

### Comment · Area_Chair_D9sF · 2023-08-14
**Additional question**

Dear Authors, Thank you for all your answers aiming to clarify the raised points so far.

I would like to encourage the reviewers who have not done so yet to read your answers and let us know.

Dear Authors, could you please also comment on how the local notion of optimality you use and for which your proof works translate (or not) into globally optimal performance? E.g. it is clear that, as you point out, the message-passing algorithms that exist for the cSBM are designed for large-dimensional features, while the present paper considered finite-dimensional ones. However, these algorithms can still be run for any dimension, and this would have been a very useful numerical comparison.

Sincerely

The Area Chair

---

> ### Author Response · Authors · 2023-08-19
> **Response to AC**
>
> > Dear Authors, could you please also comment on how the local notion of optimality you use and for which your proof works translate (or not) into globally optimal performance? E.g. it is clear that, as you point out, the message-passing algorithms that exist for the cSBM are designed for large-dimensional features, while the present paper considered finite-dimensional ones. However, these algorithms can still be run for any dimension, and this would have been a very useful numerical comparison.
>
> Thank you for the question! Let us clarify that the notion of locality refers to the local-weak topology and not the metric we use to show optimality. Empirical per-node risks can be interpreted as expected risk for a uniform at random node, making this a global notion of optimality as studied in the broader machine learning literature. Based on the example in the comment, we believe that for translation to global optimality, the AC means to ask how the architecture performs in the high-dimensional setting compared to the existing AMP algorithm in that regime, because our analysis is for the fixed-$d$ regime.
>
> Please see our general response for details on new experiments we conducted based on the questions and comments. We hope that we are able to answer the queries satisfactorily. Please let us know if there are further questions.

---

### Author Response · Authors · 2023-08-19
**General comment: Comparison with AMP algorithm on cSBM**

As suggested in the comments by the AC and some reviewers, we performed the following experiments by implementing the AMP algorithm in [1]. For our architecture, we use the $5$-hop neighbourhood for each node to gather messages, and for AMP we do $20$ iterations of message-passing. We observe the following:
1. For n=1000, d=5 (fixed-dimension setting) we observe that **our architecture performs better** than the AMP algorithm, especially when the graph signal is stronger.
2. For n=1000, d=500 (high-dimensional setting, d scaling with n), the **AMP algorithm performs better** than our architecture.

This is expected based on the theoretical results in both our paper and [1]. We expect similar results for settings with larger $n$ and $d$ in the right proportion. We post the results in a table below (since we cannot attach or link the plots), and will include the plots in the revision supplementary.

On the x-axis, we have the graph signal (first column in the table), and the y-axis is the averaged accuracy over $50$ trials for our $5$-local bayes optimal architecture and the AMP algorithm.

### For $n=1000$, $d=5$

|Graph signal| Acc% ($5$-local bayes opt) | Acc% (AMP) |
|---|--------------|----------------|
|0.30|55.2|57.9|
|0.40|**58.7**|58.4|
|0.50|**82.3**|75.6|
|0.55|**95.8**|81.1|
|0.60|**99.7**|88.0|
|0.65|**99.6**|91.3|
|0.70|**99.7**|91.1|

### For $n=1000$, $d=500$

|Graph signal| Acc% ($5$-local bayes opt) | Acc% (AMP) |
|---|--------------|----------------|
|0.30|68.4|52.3|
|0.40|72.6|66.4|
|0.50|78.1|**80.4**|
|0.55|79.7|**92.1**|
|0.60|81.8|**99.8**|
|0.65|82.3|**99.9**|
|0.70|82.5|**99.9**|

[1] Contextual Stochastic Block Models. Yash Deshpande, Subhabrata Sen, Andrea Montanari, Elchanan Mossel. Advances in Neural Information Processing Systems 31 (NeurIPS 2018).

---

> ### Comment · Reviewer_mQCx · 2023-08-19
> **Re: Comparison with AMP algorithm on cSBM**
>
> Thank you for these interesting results. However, I would take them with a pinch of salt because I was not able to reproduce them, as to the accuracy of AMP. I assumed an average degree $c=(a+b)/2=5$, feature snr $\gamma=1$, training ratio $\rho=1/2$ (semi-supervised setting, proportion of the nodes are revealed, similar to train and test on two equal-size instances); I got:
>
> ### $d=5$
>
> | $\Gamma$   | accuracy of AMP |
> | ----------- | ----------- |
> | 0.3  | 0.86       |
> | 0.4  | 0.90       |
> | 0.5  | 0.92       |
> | 0.6  | 0.95       |
>
> ---
> ### $d=500$
>
> | $\Gamma$   | accuracy of AMP |
> | ----------- | ----------- |
> | 0.3  | 0.83       |
> | 0.4  | 0.87       |
> | 0.5  | 0.90       |
> | 0.6  | 0.94       |
>
> (one percent of std). I guess the values of the parameters are not the good ones. For instance, I obtain an accuracy of 99.9% too at $d=5, \Gamma=0.6$ if the average degree is $c=20$; maybe the authors could precise what values they used.
>
> Also, maybe the authors did not implement the right algorithm. In [1] the parameterization differ from their article. They should consider the algorithm given part 6 and modify it slightly to incorporate the train labels, by fixing the marginals of the train nodes to their true values (as I explained earlier in my comment W4).

---

> > ### Author Response · Authors · 2023-08-20
> > **Additional response**
> >
> > It is unfortunate that we cannot share our code and plots in the discussion period for an exact reproduction.
> > We agree that, between this lack of transparency and the time constraints, simulations presented during the discussion period should be taken with a grain of salt.
> > Perhaps we can clarify a bit more about our simulations to aid in the comparison with the reviewer's comment.
> >
> > We used the AMP algorithm from [1].  The parameterization in [1] is indeed different from our paper and we carefully checked this to ensure that our implementation is correct to the best of our knowledge. Unfortunately, we were unable to compare to the modified AMP algorithm suggested by the reviewer as we are not clear precisely which modifications the reviewer wishes us to perform. Nevertheless, we hope these simulations address the reviewer's concerns.
> >
> > Here are the parameterization details of our experiments.
> >
> > ---
> > ### Parameters for our initial experiments above
> > - $n=1000$ nodes, $d=5$ for the fixed-$d$ setting.
> > - Feature SNR fixed to $1$.
> > - $a=20$ with varying $b$ for a range of values corresponding to $\Gamma=\frac{|a-b|}{a+b}\in[0.3, 0.8]$.
> > - $\ell=5$ for our architecture, trained using half of the nodes.
> > - $t=20$ iterations for the AMP algorithm.
> > ---
> >
> > We repeated our experiments with the reviewer's choice of parameters for $n=1000,d=5$ and $t\in\\\{5,20\\\}$ iterations of AMP. We obtained the following result which seems to roughly match (for $t=20$) what the reviewer has in their comment. The reported accuracies are averaged over $10$ trial runs.
> >
> > |Graph signal| Acc ($5$-local bayes opt) | Acc (AMP, $t=5$) | Acc (AMP, $t=20$) |
> > |---|--------------|----------------|--|
> > |0.3|0.870|0.753|0.858|
> > |0.4|0.954|0.816|0.890|
> > |0.5|0.988|0.819|0.916|
> > |0.6|0.996|0.892|0.940|
> >
> > We should note however that these experiments might be apples-to-oranges comparisons: We are comparing a $5$-local algorithm to a $20$-iteration AMP.
> > Naively, however, it seems more natural to compare AMP with $t$ iterations to a $t$-local algorithm. In light of this, we have included $t=5$ iterations of AMP in the table above. As seen, $t=5$ iterations of AMP does not perform well. (This is why we chose $\ell=5$ and $t=20$ for our initial experiments.)

---

### Decision · Program_Chairs · 2023-09-21

**Decision:**

Accept (poster)

**Comment:**

This paper was considered to bring an interesting technical contribution. The discussion clarified many of the referees' questions, and the authors should carry those answers into the manuscript. In particular, it should be made very clear under what conditions the claims of optimality hold and present explicit examples of cases (via numerical experiments) where the proposed architecture is not optimal (e.g. high-dimensional features). Without this, the paper may mislead the reader that the results are stronger than what they are. The results are considered interesting, and the limitations should be put forward already in the introduction rather than being hidden in mathematical formulations of the limiting assumptions in the later technical pages.